# Evaluating Representation Learning on the Protein Structure Universe

**Arian R. Jamasb**[*,1,†], **Alex Morehead**[*,2], **Chaitanya K. Joshi**[*,1], **Zuobai Zhang**[*,3],
**Kieran Didi**[1], **Simon Mathis**[1], **Charles Harris**[1], **Jian Tang**[3], **Jianlin Cheng**[2],
**Pietro Liò**[1], **Tom L. Blundell**[1]

[1]University of Cambridge,    [2]University of Missouri,    [3]Mila - Québec AI Institute

## Abstract

We introduce *ProteinWorkshop*, a comprehensive benchmark suite for representation learning on protein structures with Geometric Graph Neural Networks. We consider large-scale pre-training and downstream tasks on both experimental and predicted structures to enable the systematic evaluation of the quality of the learned structural representation and their usefulness in capturing functional relationships for downstream tasks. We find that: (1) large-scale pretraining on AlphaFold structures and auxiliary tasks consistently improve the performance of both rotation-invariant and equivariant GNNs, and (2) more expressive equivariant GNNs benefit from pretraining to a greater extent compared to invariant models.

We aim to establish a common ground for the machine learning and computational biology communities to rigorously compare and advance protein structure representation learning. Our open-source codebase reduces the barrier to entry for working with large protein structure datasets by providing: (1) storage-efficient dataloaders for large-scale structural databases including AlphaFoldDB and ESM Atlas, as well as (2) utilities for constructing new tasks from the entire PDB. *ProteinWorkshop* is available at: `github.com/a-r-j/ProteinWorkshop`.

## 1 Introduction

Modern protein structure prediction methods have led to an explosion in the availability of structural data (Jumper et al., 2021; Baek et al., 2021). While many sequence-based functional annotations can be directly mapped to structures, this has resulted in a significantly-increasing gap between structures and meaningful *structural* annotations (Varadi et al., 2021). Recent work has focused on developing methods to draw biological insights from large volumes of structural data by either determining representative structures that can be used to provide such annotations (Holm, 2022) or representing structures in a simplified and compact manner such as sequence alphabets (van Kempen et al., 2023) or graph embeddings (Greener & Jamali, 2022). These works have significantly reduced the computational resources required to process and analyse such structural representatives at scale. Nonetheless, it remains to be shown how such results can help us better understand the relationship between protein sequence, structure, and function through the use of deep learning algorithms.

Several deep learning methods have been developed for protein structures. In particular, Geometric Graph Neural Networks (GNNs) (Duval et al., 2023) have emerged as the architecture of choice for learning structural representations of biomolecules (Schütt et al., 2018; Gasteiger et al., 2020; Jing et al., 2020; Schütt et al., 2021; Morehead et al., 2022; Zhang et al., 2023b). Methods can be categorised according to (1) the featurisation schemes and level of granularity of input structures ($C\alpha$, backbones, all-atom); as well as (2) the enforcement of physical symmetries and inductive biases (invariant or equivariant representations) (Joshi et al., 2023a). However, there remains a need for a robust, standardised benchmark to track the progress of new and established methods with greater granularity and relevance to downstream applications.

---

[*]Equal contribution
[†]Current affiliation: Prescient Design, Genentech

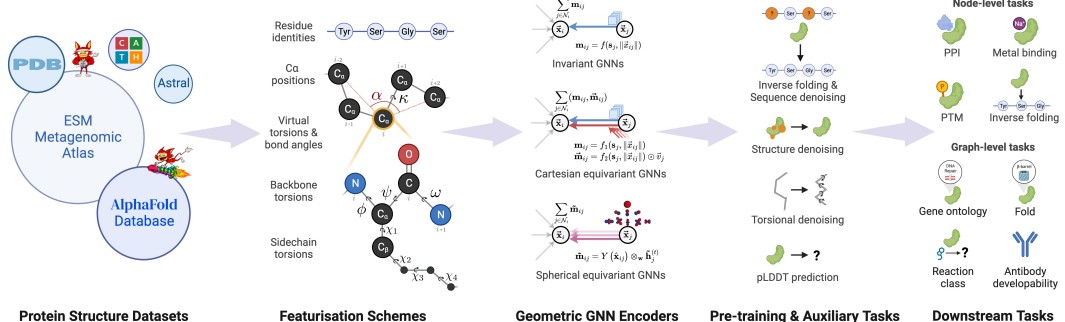

Figure 1: Overview of *ProteinWorkshop*, a comprehensive benchmark suite for evaluating pre-training and representation learning of Geometric GNNs on large-scale protein structure data.

In this work, we develop a unified and rigorous framework for evaluating protein structural encoders, providing pretraining corpora that span known foldspace and tasks that assess the ability of models to learn informative representations at different levels of structural granularity. Previous works in protein structure representation learning have focused on learning effective *global* (i.e. graph-level) representations of protein structure, typically evaluating the methods on function or fold classification tasks (Gligorijević et al., 2021; Zhang et al., 2023b). However, there has been comparatively little investigation into the ability of different methods to learn informative local (*node-level*) representations. Good node-level representations are important for a variety of annotation tasks, such as binding or interaction site prediction (Gainza et al., 2020), as well as providing conditioning signals in structure-conditioned molecule design methods (Schneuing et al., 2022; Corso et al., 2023). Understanding the structure-function relationship at this granular level can drive progress in protein design by revealing structural motifs that underlie desirable properties, enabling them to be incorporated into designs.

Our contributions are as follows:

- We curate numerous *structure-based* pretraining and fine-tuning datasets from the literature with a focus on tasks that can enable structural annotation of predicted structures. We compile a highly-modular benchmark, enabling the community to rapidly evaluate protein representation learning methods across tasks, models, and pretraining setups.

- We benchmark Geometric GNNs for representation learning of proteins at different levels of structural granularity (C$\alpha$, backbones, sidechain) and across several classes of models, ranging from general-purpose (Schütt et al., 2018; Satorras et al., 2021) to protein-specific architectures (Morehead & Cheng, 2024; Zhang et al., 2023b). We are the first to evaluate higher order equivariant GNNs (Thomas et al., 2018; Batatia et al., 2022) for proteins.

- We pretrain and evaluate models on, to our knowledge, the *largest non-redundant* protein structure corpus containing 2.27 million structures from AlphaFoldDB.

- Our benchmarks show that sequence and structure denoising-based auxiliary tasks and structure denoising-based pretraining consistently improve Geometric GNNs. Moreover, we surprisingly observe that sequence-based pretrained ESM-2-650M (Lin et al., 2022) augmented with our structural featurisation matches state-of-the-art GNNs on (super)family fold and gene ontology prediction.

## 2 PROTEINWORKSHOP

The overarching goal of *ProteinWorkshop* is to effectively cover the design space of protein structure representation learning methods. To achieve this, the benchmark is highly modular by design, enabling evaluation of different combinations of structural encoders, protein featurisation schemes, and auxiliary tasks over a wide range of both supervised and unsupervised tasks. A user manual is available in Appendix D, containing detailed listings and descriptions of all components.

## 2.1 Featurisation Schemes

Protein structures are typically represented as geometric graphs, with researchers opting to use a coarse-grained C$\alpha$ atoms graph as full atom representations can quickly become computationally intractable due to a large number of nodes. However, this is a lossy representation, with much of the structural detail, such as orientation of the backbone and sidechain structure, being only implicitly encoded. Due to the computational burden incurred by operating on full-atom node representations, we focus primarily on C$\alpha$-based graph representations, investigating featurisation strategies to incorporate higher-level structural information. Note that we do provide utilities to enable users to work with backbone and full-atom graphs in the benchmark. Details about different featurisation schemes are provided in Appendix D.4 and Table 5.

## 2.2 Pre-training Tasks

The benchmark contains a comprehensive suite of pretraining tasks. Broadly, these can be categorised into: masked-attribute prediction, denoising-based and contrastive learning-based tasks. These can be used as both a pretraining objective or as auxiliary tasks in a downstream supervised task.

**Sequence Denoising.** The benchmark contains two variations based on two sequence corruption processes $C(\tilde{\mathcal{S}}|\mathcal{S}, \nu)$ that receive an amino acid sequence $\mathcal{S} \in [0, 1]^{|\mathcal{V}| \times 23}$ and return a sequence $\mathcal{S} \in [0, 1]^{|\mathcal{V}| \times 23}$ with fraction $\nu$ of its positions corrupted. The first scheme is based on mutating a fraction of the residues to a random amino acid and tasking the model with recovering the uncorrupted sequence. The second is a masked residue prediction task, where a fraction of the residues are altered to a mask value and the model is tasked to recover the uncorrupted sequence.

**Structure Denoising.** We provide two structure-based denoising tasks: coordinate denoising and torsional denoising. In the coordinate denoising task, noise is sampled from a normal or uniform distribution and scaled by noise factor, $\nu \in \mathbb{R}$, and applied to each of the atom coordinates in the structure to ensure structural features, such as backbone or sidechain torsion angles, are also corrupted. The model is then tasked with predicting either the per-node noise or the original uncorrupted coordinates. For torsional denoising, the noise is applied to the backbone torsion angles and Cartesian coordinates are recomputed using pNeRF (AlQuraishi, 2019) and the uncorrupted bond lengths and angles prior to feature computation. Similarly to the coordinate denoising task, the model is then tasked with predicting either the per-residue angular noise or the original dihedral angles.

**Sequence-Structure Co-Denoising.** This is a multitask formulation of the previously described structure and sequence denoising tasks, with separate output heads for denoising each modality.

**Masked Attribute Prediction.** We use inverse folding (Section 2.3.1) as a pretraining task. The benchmark additionally incorporates the distance, angle and dihedral angle masked-attribute prediction (Zhang et al., 2023b) as well as a backbone dihedral angle prediction task.

**pLDDT Prediction.** Structure prediction models typically provide per-residue pLDDT (predicted Local Distance Difference Test) scores as local confidence measures which have been shown to correlate well with disordered regions (Wilson et al., 2022). We formulate a node-level regression task on predicted structures, somewhat analogous to structure quality assessment, where the model is tasked with predicting the scaled per-residue pLDDT $y \in [0, 1]$ values.

## 2.3 Downstream Tasks

We curate several structure-based and sequence-based datasets from the literature and existing benchmarks[†], summarised in Table 1. The tasks are selected to evaluate not only the *global* structure representational power of each method, but also to evaluate the ability of each method to learn informative *local* representations for residue-level prediction and annotation tasks.

The raw structures are, where possible and accounting for obsolescence, retrieved directly from the PDB (or another structural source) as several processed datasets used by the community discard full atomic coordinates in favour of retaining only $C_\alpha$ positions, making them unsuitable for in-depth

---

[†]To retain focus on *protein* representation learning, we deliberately exclude commonly-used tasks based on protein-small molecule interactions as it is hard to disentangle the effect of the small molecule representation and the potential for bias (Boyles et al., 2019).

Table 1: **Overview of supervised tasks and datasets.**

| | Task | Dataset Origin | Structures | # Train | # Validation | # Test | Metric |
|---|---|---|---|---|---|---|---|
| Node-level | Inverse Folding | Ingraham et al. (2019) | Experimental | 3.9 M | 105 K | 180 K | Perplexity |
| | PPI Site Prediction | Gainza et al. (2020) | Experimental | 478 K | 53 K | 117 K | AUPRC |
| | Metal Bind Site Prediction | | Experimental | 1.1 M | 13.7 K | 29.8 K | Accuracy |
| | PTM Site Prediction | Yan et al. (2023) | Predicted | 44 K | 2.4 K | 2.5 K | ROC-AUC |
| Graph-level | Fold Prediction | Hou et al. (2017) | Experimental | 12.3 K | 0.7 K | 1.3/0.7/1.3 K | Accuracy |
| | Gene Ontology Prediction | Gligorijević et al. (2021) | Experimental | 27.5 K | 3.1 K | 3.0 K | $F_{max}$ |
| | Reaction Class Prediction | Hermosilla et al. (2020) | Experimental | 29.2 K | 2.6 K | 5.6 K | Accuracy |
| | Antibody Dev. Prediction | Huang et al. (2021) | Experimental | 1.7 K | 0.24 K | 0.48 K | AUPRC |

experimentation. This provides an entry point for users to apply a custom sequence of pre-processing steps, such as deprotonation or fixing missing regions which are common in experimental data.

### 2.3.1 NODE-LEVEL TASKS

**Inverse Folding.** Many generative methods for protein design produce backbone structures that require the design of an associated sequence. As a result, inverse folding is an important part of *de novo* design pipelines for proteins (Dauparas et al., 2022). Formally, this is a node-level classification task where the model learns a mapping for each residue $r_i$ to an amino acid type $y \in \{1, \ldots, n\}$, where $n$ is the vocabulary size ($n = 20$ for the canonical set of amino acids). Inverse folding is a generic task that can be applied to any dataset in the benchmark. In the literature, it is commonly evaluated on the CATH dataset (Section 2.4) compiled by Ingraham et al. (2019).

**PPI Site Prediction.** Identifying protein-protein interaction sites has important applications in developing refined protein-protein interaction networks and docking tools, providing biological context to guide protein engineering and target identification in drug discovery campaigns (Jamasb et al., 2021). This task is a node-level binary classification task where the goal is to predict whether or not a residue is involved in a protein-protein interaction interface. We use the dataset of experimental structures curated from the PDB by Gainza et al. (2020) and retain the original splits, though we modify the labelling scheme to be based on inter-atomic proximity (3.5 Å), which can be user-defined, rather than solvent exclusion. The dataset is curated from the PDB by preprocessing such as the presence of one of the seven specified ligands (e.g., ADP or FAD), clustering based on 30% sequence identity and random subsampling. It contains 1,459 structures, which are randomly assigned to training (72%), validation (8%) and test set (20%). 12 (Å) radius patches were extracted from the generated structures and a patch labelled as part of a binding pocket if its centre point was < 3 (Å) away from an atom of the corresponding ligand.

**Metal Binding Site Prediction.** Many proteins coordinate transition metal ions to carry out their functions. Predicting the binding sites of metal ions can elucidate the role of metal binding on protein function. This is a binary node classification task where each residue is mapped to a label $y \in \{0, 1\}$ indicating whether the residue (or its constituent atoms) is within 3.5 (Å) of a user-defined metal ion or ligand heteroatom, respectively. We provide tooling to curate a dataset of experimental structures from the PDB for this task, where binding site assignments for each residue are computed on-the-fly. While the benchmark supports this task on arbitrary subsets of the PDB and ligands, we provide the Zinc-binding dataset from Dürr et al. (2023) specifically for this task. The dataset is constructed by sequence-based clustering of the PDB at 30% sequence identity to remove sequence and structural redundancy. Clusters with a member shorter than 3000 residues, containing at least one zinc atom with resolution better than 2.5 (Å) determined by x-ray crystallography and not containing nucleic acids are used to compose the dataset. If multiple structures fulfil these criteria, the highest resolution structure is used. The train (2,085) / validation (26) / test (59) splits are constructed such that proteins in the validation and test sets have no partial overlap with any protein in the training data.

**Post-Translational Modification Site Prediction.** Identifying the precise sites where post-translational modifications (PTMs) occur is essential for understanding protein behaviour and designing targeted therapeutic interventions. We frame prediction of PTM sites as a multilabel classification task where each residue is mapped to a label $y \in \{1, \ldots, 13\}$ distinguishing between modifications on different amino acids (e.g. phosphorylation on S/T/Y and N-linked glycosylation on N). We use a dataset of 48,811 AlphaFold2-predicted structures curated by Yan et al. (2023),

where each structure contains the PTM metadata necessary to construct residue-wise site prediction labels. The dataset is split into training (43,907, validation (2,393) and test (2,511) sets based on 50% sequence identity and 80% coverage.

### 2.3.2 GRAPH-LEVEL TASKS

**Fold Prediction.** The utility of this task is that it serves as a litmus test for the ability of a model to distinguish different structural folds. It stands to reason that models that perform poorly on distinguishing fold classes likely learn limited or low-quality structural representations. This is a multiclass graph classification task where each protein, $\mathcal{G}$, is mapped to a label $y \in \{1, \ldots, 1195\}$ denoting the fold class. We adopt the fold classification dataset originally curated from SCOP 1.75 by (Hou et al., 2017). This provides three different test sets stratified based on topological similarity: Fold, in which proteins originating from the same superfamily are absent during training; Superfamily, in which proteins originating from the same family are absent during training; and Family, in which proteins from the same family are present during training.

**Gene Ontology Prediction.** Predicting protein function in the form of functional annotations such as GO terms has important applications in protein analysis and engineering, providing researchers with the ability to cluster functionally-related structures or to guide protein generation methods to design new proteins with desired functional properties. This is a multilabel classification task, assigning functional Gene Ontology (GO) annotation to structures. GO annotations are assigned within three ontologies: biological process (BP), cellular component (CC) and molecular function (MF). We use the dataset of experimental structures curated from the PDB by Gligorijević et al. (2021) and retain the original multi-cutoff based splits, with cutoff at 30% sequence similarity.

**Reaction Class Prediction.** As proteins' reaction classifications are based on their enzyme-catalyzed reaction according to all four levels of the standard Enzyme Commission (EC) number, methods that predict such classifications may help elucidate the function of newly-designed proteins as they are developed. This is a multiclass graph classification task where each protein, $\mathcal{G}$, is mapped to a label $y \in \{1, ..., 384\}$ denoting which class of reactions a given protein catalyzes; all four levels of the EC assignment are employed to define the reaction class label. We adopt the reaction class prediction dataset originally curated from the PDB by Hermosilla et al. (2020), split on the basis of sequence similarity using a 50% threshold.

**Antibody Developability Prediction.** Therapeutic antibodies must be optimised for favourable physicochemical properties in addition to target binding affinity and specificity to be viable development candidates. Consequently, we frame prediction of antibody developability as a binary graph classification task indicating whether a given antibody is developable. We adopt the antibody developability dataset originally curated from SabDab (Dunbar et al., 2014) by Chen et al. (2020). This dataset contains 2,426 antibodies that have both sequences and PDB structures available, where each example contains both a heavy chain and a light chain with resolution $< 3$ (Å). The label is based on thresholding the developability index (DI) (Lauer et al., 2012) as computed by BIOVIA's platform (Systèmes, 2016), which relies on an antibody's hydrophobic and electrostatic interactions. This task is interesting from a benchmarking perspective as it enables targeted performance assessment of models on a specific (immunoglobulin) fold, providing insight into whether general-purpose structure-based encoders can be applicable to fold-specific tasks.

### 2.4 PRE-TRAINING DATASETS

The benchmark contains several large corpora of both experimental and predicted structures that can be used for pretraining or inference. We provide utilities for configuring supervised tasks and splits directly from the PDB. Additionally, we build storage-efficient dataloaders for large pretraining corpora of predicted structures (AlphaFoldDB, ESM Atlas). We believe our codebase will considerably reduce the barrier to entry for working with large structure-based datasets.

### 2.4.1 EXPERIMENTAL STRUCTURES

**PDB.** We provide utilities for curating datasets directly from the Protein Data Bank (Berman, 2000). In addition to using the collection in its entirety, users can define filters to subset and split the data using a combination of structural similarity, sequence similarity or temporal strategies. Structures can

be filtered by length, number of chains, resolution, deposition date, presence/absence of particular ligands and structure determination method.

**CATH.** We provide the dataset derived from CATH 4.2 40% (Knudsen & Wiuf, 2010) non-redundant chains developed by Ingraham et al. (2019) as an additional, smaller, pretraining dataset.

**ASTRAL.** ASTRAL (Brenner, 2000) provides protein *domain* structures which are regions of proteins that can maintain their structure and function independently of the rest of the protein. Domains typically exhibit highly-specific functions and can be considered structural building blocks.

### 2.4.2 PREDICTED STRUCTURES

**AlphaFoldDB Representative Structures.** This dataset contains 2.27 million representative structures, identified through large-scale structural-similarity-based clustering of the 214 million structures contained in the AlphaFold Database (Varadi et al., 2021) using FoldSeek (van Kempen et al., 2023). We additionally provide a subset of this collection — the so-called dark proteome — corresponding to the 31% of the representative structures that lack annotations.

**ESM Atlas, ESM High Quality.** These datasets are compressed collections of predicted structures produced by ESMFold. ESM Atlas is the full collection of all 772m predicted structures for the MGnify 2023 release (Richardson et al., 2022). ESM High Quality is a curated subset of high confidence (mean pLDDT) structures from the collection.

## 3 METHODS AND EXPERIMENTAL SETUP

**Overview.** To demonstrate the utility of our benchmark, we investigate how combinations of protein structure representation, architecture choice and pretraining/auxiliary tasks affect predictive performance across a range of tasks. The tasks are selected to focus on important real-world structural annotation tasks and such that we can evaluate these combinations in terms of both the local and global representational power. To this end, we select state-of-the-art protein structure encoders and generic geometric GNN architectures that span the design space of geometric GNN models with regard to both message passing body order and tensor order (Joshi et al., 2023a). We evaluate several structural representations that, to varying degrees, capture the full detail of the protein structure.

**Architectures.** We provide a unified implementation of several rotation invariant and equivariant architectures. We benchmark 4 general purpose models: SchNet (Schütt et al., 2018), EGNN (Satorras et al., 2021), TFN (Thomas et al., 2018), MACE (Batatia et al., 2022); and 2 protein-specific architectures: GCPNet (Morehead & Cheng, 2024), GearNet (Zhang et al., 2023b). We also compare geometric GNNs to the pretrained sequence-based language model ESM (Lin et al., 2022) augmented with structural featurisation. We chose the 650M pretrained ESM-2 because this is the scale at which significant structure-related abilities were observed for ESM.

**Featurisation Schemes.** We consider five featurisation schemes, progressively increasing the amount of structural information provided to the model by incorporating sequence positional information, virtual dihedral and bond angles over the $C\alpha$ trace, backbone torsion angles, and sidechain torsion angles. Featurisation schemes are detailed in Table 5 in the Appendix.

**Pretraining Dataset.** For all pretraining experiments we use AlphaFoldDB (Barrio-Hernandez et al., 2023). This dataset provides a rich diversity of 2.27 million non-redundant protein structures and, to our knowledge, is substantially more diverse than any other previously used structure-based pretraining corpus, whilst remaining of a size that is amenable to experimentation. Models pretrained on AlphaFoldDB should, in principle, exhibit strong generalisation to the currently known (and predicted) natural protein structure universe as it would have 'seen' the same protein fold during pretraining. To facilitate working with large-scale AlphaFoldDB and ESM Atlas, we developed storage-efficient dataloaders based on FoldComp (Kim et al., 2023), described in Appendix D.6.

**Pretraining and Auxiliary Tasks.** In our evaluation, we focus predominantly on denoising-based pretraining and auxiliary tasks as these are comparatively less explored than contrastive or masked-attribute prediction tasks (Zhang et al., 2023b). We consider five pretraining tasks: (1) structure-based denoising, (2) sequence denoising, (3) torsional denoising, (4) inverse folding and (5) pLDDT prediction. Structure and sequence denoising are also used as auxiliary tasks in our experiments. We

also investigate an inverse folding pre-training task which we subsequently finetune on the CATH dataset for benchmarking inverse folding as a downstream task (see below).

**Noising Schemes.** For structure-based denoising we draw i.i.d. noise samples from a Gaussian distribution and scale by $\sigma = 0.1$ to corrupt the input coordinates or dihedral angles. Geometric scalar and vector-valued features are computed from the noised structure, *i.e.* $\tilde{\mathcal{G}} = (\mathcal{V}, \tilde{\mathcal{E}}, \tilde{\mathbf{X}}, \tilde{\mathbf{S}}, \tilde{\mathbf{V}})$, where $\tilde{\boldsymbol{x}}_i = \boldsymbol{x}_i + \sigma \boldsymbol{\epsilon}_i$ and $\boldsymbol{\epsilon}_i \sim \mathcal{N}(0, I_3)$. For sequence-based denoising, we use the mutation strategy and corrupt 25% of the residues in each protein. When sequence denoising is used as an auxiliary task, we weight the loss with a coefficient $\lambda = 0.1$, similar to NoisyNodes (Godwin et al., 2021).

**Training.** As we are interested in benchmarking large-scale datasets and models, we try to consistently use six layers for all models, each with 512 hidden channels. For equivariant GNNs, we reduced the number of layers and hidden channels to fit 80GB of GPU memory on one NVIDIA A100 GPU. For downstream tasks, we set the maximum number of epochs to 150 and use the Adam optimizer with a batch size of 32 and ReduceLROnPlateau learning rate scheduler, monitoring the validation metric with patience of 5 epochs and reduction of 0.6. See Appendix D.10 for details on hyperparameter tuning for optimal learning rates and dropouts for each architecture. We train models to convergence, monitoring the validation metric and performing early stopping with a patience of 10 epochs. Pretraining is performed for 10 epochs using a linear warm-up with cosine schedule. We report standard deviations over three runs across three random seeds.

## 4 RESULTS & DISCUSSIONS

### 4.1 AUXILIARY DENOISING CONSISTENTLY IMPROVES BASELINE PERFORMANCE

In Table 2, we first set out to determine the following questions about architectural choices in conjunction with denoising auxiliary tasks and *without* pretraining:

1. **Whether invariant or equivariant models perform better?** Across 10 tasks, equivariant models such as EGNN and GCPNet attain the best performance on 5. Notably, sequence-based ESM-2-650M augmented with our structural featurisation matches state-of-the-art protein-specific GNNs (Fan et al., 2023) on (super)family and gene ontology prediction.

2. **Which input representation is the best for each respective task?** Featurising models with $C\alpha$ atoms, virtual angles, and backbone torsions provides the best performance overall on 22 out of 60 combinations of models and tasks. This suggests that letting models implicitly learn about side chain orientation and flexibility by using backbone-only featurisation may prevent overfitting on crystallisation artifacts (Dauparas et al., 2022).

3. **Whether auxiliary denoising tasks improve model performance?** Both sequence and structure denoising are particularly useful auxiliary tasks for training protein structure encoders, until sufficient structural detail makes the tasks trivial, improving results over not using auxiliary tasks for 50 out of 60 combinations of models and primary tasks. Notably, structure denoising helped stabilise the training of MACE models on the GO and Reaction tasks, where other runs did not converge.

### 4.2 INCORPORATING MORE STRUCTURAL DETAIL IMPROVES PRE-TRAINING PERFORMANCE

We then investigated protein structure pre-training in Table 3 to determine:

1. **Which input representation is best for pre-training?** Incorporating greater structural detail with dihedral angles generally improves validation metrics on pre-training tasks, more so than architecture.

2. **Which GNNs benefit from which pre-training task?** Inverse folding, structure denoising, sequence denoising, and torsional denoising benefit equivariant models the most in the context of pre-training, whereas pLDDT prediction benefits invariant models the most, suggesting that certain pre-training tasks benefit certain classes of models more than other tasks. Unfortunately, we were currently unable to pre-train spherical equivariant GNNs (TFN, MACE) due to the high computational requirements of these models.

Table 2: **Baseline benchmark results without pretraining.** Results for each model and featurisation pair are given as: no auxiliary task / +sequence denoising / +structure denoising . Coloured boxes mark the best auxiliary tasks per model and featurisation, underlined results denote the best featurisation choice per model, and **bold results** are the best models for each task. Greyed cells denote invalid task-setup combinations (e.g. inverse folding and sequence denoising as auxiliary task), and —— denote runs that did not converge. [*] denotes results for ESM-MSA-1b without structural features, taken from Zhang et al. (2023b), whereas [†] denotes results for new tasks with ESM-2-650M. **Key takeaways:** (1) Auxiliary tasks consistently improve performance across models and primary tasks compared to performance without auxiliary tasks. (2) Equivariant GNNs outperform invariant GNNs in general though protein-specific architectures are consistently performant. (3) $C\alpha$, virtual angles, and backbone torsions provide the best featurisation. (4) Augmenting ESM-2-650M with structural features provides compelling performance for (super)family fold classification and gene ontology prediction compared to GNNs. For enhanced readability, in Section A of the Appendix, we provide a series of analysis figures characterizing each task's results.

| Method | Features | GO-BP (↑) | GO-MF (↑) | GO-CC (↑) | Ab. Dev. (↑) | Fold (↑) — Fold | Fold (↑) — Family | Fold (↑) — Superfamily | Reaction (↑) | PPI (↑) | Inverse Folding (↓) |
|---|---|---|---|---|---|---|---|---|---|---|---|
| ESM | Seq. | 0.462* | 0.546* | 0.394* | 0.908±0.01† | 26.8* | 97.8* | 60.1* | 83.1* | 0.955±0.00 † | N/A |
| | + κ,α,φ,ψ,ω | **0.472**±0.00 | **0.583**±0.00 | **0.545**±0.00 | 0.885±0.00 | 34.59±0.00 | **99.33**±0.00 | **72.71**±0.00 | 82.11±0.00 | 0.956±0.00 | N/A |
| SchNet | Cα | 0.314±0.00 / 0.314±0.02 / 0.343±0.00 | 0.365±0.00 / 0.359±0.4 / 0.408±0.01 | 0.387±0.01 / 0.415±0.00 / 0.418±0.00 | 0.872±0.01 / 0.878±0.01 / 0.872±0.01 | 20.71±0.01 / 24.58±0.01 / 19.95±0.00 | 76.70±0.02 / 80.25±0.01 / 73.45±0.03 | 23.75±0.01 / 28.65±0.01 / 22.14±0.02 | 58.94±0.01 / 68.33±0.01 / 57.59±0.03 | 0.954±0.00 / 0.952±0.01 / 0.955±0.00 | |
| | + Seq. | 0.312±0.01 / 0.257±0.01 / 0.285±0.01 | 0.369±0.03 / 0.283±0.07 / 0.409±0.00 | 0.390±0.00 / 0.374±0.03 / 0.421±0.00 | 0.875±0.01 / 0.889±0.01 / 0.876±0.01 | 28.14±0.00 / 31.98±0.01 / 27.58±0.01 | 87.75±0.02 / 89.51±0.00 / 88.43±0.03 | 32.88±0.02 / 37.56±0.01 / 32.80±0.01 | 70.30±0.01 / 68.83±0.01 / 62.94±0.10 | 0.953±0.00 / 0.950±0.00 / 0.953±0.00 | 11.78±0.08 / 12.34±0.11 |
| | + κ,α | 0.322±0.03 / 0.264±0.01 / 0.335±0.03 | 0.410±0.01 / 0.273±0.01 / 0.417±0.01 | 0.403±0.00 / 0.379±0.01 / 0.420±0.00 | 0.871±0.01 / 0.896±0.00 / 0.867±0.02 | 27.20±0.02 / 30.81±0.01 / 29.83±0.01 | 87.97±0.04 / 91.05±0.01 / 90.22±0.01 | 33.29±0.03 / 39.38±0.01 / 32.87±0.02 | 70.33±0.01 / 72.58±0.01 / 65.93±0.03 | 0.953±0.00 / 0.951±0.00 / 0.954±0.00 | 11.03±0.03 / 11.81±0.50 |
| | + φ,ψ,ω | 0.355±0.00 / 0.308±0.00 / 0.366±0.00 | 0.422±0.00 / 0.362±0.03 / 0.444±0.01 | 0.412±0.01 / 0.406±0.01 / 0.429±0.00 | 0.861±0.02 / 0.878±0.01 / 0.848±0.01 | 29.89±0.01 / 29.48±0.01 / 26.90±0.02 | 90.58±0.01 / 90.55±0.01 / 90.14±0.03 | 36.18±0.02 / 37.66±0.01 / 34.34±0.04 | 67.53±0.03 / 73.83±0.02 / 72.28±0.01 | 0.954±0.00 / 0.952±0.00 / 0.956±0.00 | 9.97±0.09 / 10.76±0.02 |
| | + χ₁₋₄ | 0.334±0.00 / 0.266±0.02 / 0.356±0.00 | 0.404±0.00 / 0.322±0.00 / 0.434±0.00 | 0.390±0.00 / 0.392±0.01 / 0.423±0.00 | 0.869±0.00 / 0.869±0.03 / 0.847±0.01 | 28.81±0.02 / 27.42±0.02 / 29.48±0.02 | 89.54±0.00 / 87.34±0.01 / 90.31±0.01 | 36.22±0.01 / 34.05±0.01 / 35.10±0.02 | 68.72±0.03 / 68.87±0.00 / 71.40±0.01 | 0.954±0.00 / 0.951±0.00 / 0.955±0.00 | |
| GearNet-Edge | Cα | 0.393±0.00 / 0.394±0.00 / 0.404±0.00 | 0.474±0.01 / 0.467±0.00 / 0.478±0.00 | 0.450±0.00 / 0.408±0.07 / 0.453±0.00 | 0.807±0.04 / 0.817±0.05 / 0.818±0.02 | 30.90±0.02 / 30.81±0.02 / 31.61±0.01 | 93.40±0.02 / 94.46±0.01 / 93.68±0.04 | 44.65±0.04 / 46.78±0.04 / 46.04±0.05 | 78.14±0.01 / 80.03±0.01 / 79.12±0.0 | 0.959±0.00 / 0.950±0.00 / 0.961±0.00 | |
| | + Seq. | 0.395±0.00 / 0.398±0.00 / 0.403±0.00 | 0.475±0.00 / 0.468±0.00 / 0.483±0.01 | 0.437±0.00 / 0.439±0.01 / 0.400±0.07 | 0.815±0.01 / 0.837±0.01 / 0.815±0.03 | 33.20±0.00 / 31.02±0.02 / 32.18±0.02 | 95.33±0.00 / 94.58±0.01 / 93.20±0.02 | 46.74±0.02 / 44.20±0.01 / 45.99±0.00 | 77.02±0.03 / 77.74±0.03 / 75.80±0.03 | 0.956±0.00 / 0.956±0.00 / 0.956±0.00 | 12.79±0.17 / 12.60±0.16 |
| | + κ,α | 0.393±0.00 / 0.394±0.00 / 0.401±0.00 | 0.476±0.00 / 0.466±0.00 / 0.483±0.00 | 0.436±0.00 / 0.431±0.00 / 0.441±0.00 | 0.830±0.03 / 0.821±0.01 / 0.787±0.01 | 32.79±0.01 / 33.73±0.00 / 31.88±0.00 | 95.35±0.01 / 95.48±0.00 / 94.92±0.00 | 47.56±0.02 / 48.39±0.02 / 47.32±0.00 | 77.45±0.01 / 77.76±0.01 / 76.88±0.00 | 0.958±0.00 / 0.955±0.00 / 0.957±0.00 | 12.35±0.05 / 11.91±0.13 |
| | + φ,ψ,ω | 0.397±0.00 / 0.402±0.00 / 0.483±0.00 | 0.480±0.00 / 0.470±0.00 / 0.483±0.00 | 0.441±0.00 / 0.443±0.00 / 0.442±0.01 | 0.811±0.03 / 0.801±0.03 / 0.820±0.01 | 33.75±0.01 / 34.02±0.00 / 34.63±0.01 | 94.35±0.00 / 94.89±0.00 / 95.31±0.01 | 46.60±0.03 / 48.09±0.01 / 47.78±0.02 | 76.61±0.01 / 78.20±0.01 / 76.57±0.02 | 0.954±0.01 / 0.956±0.00 / 0.960±0.00 | 11.61±0.12 / 11.23±0.09 |
| | + χ₁₋₄ | 0.384±0.00 / 0.390±0.00 / 0.389±0.00 | 0.459±0.00 / 0.467±0.01 / 0.482±0.00 | 0.430±0.01 / 0.437±0.00 / 0.437±0.00 | 0.823±0.02 / 0.815±0.01 / 0.831±0.02 | 31.05±0.01 / 32.32±0.00 / 32.09±0.00 | 93.96±0.01 / 93.72±0.00 / 94.24±0.00 | 45.01±0.01 / 43.90±0.00 / 44.21±0.00 | 75.91±0.03 / 74.14±0.01 / 77.76±0.01 | 0.959±0.00 / 0.958±0.00 / 0.962±0.00 | |
| EGNN | Cα | 0.358±0.00 / 0.347±0.00 / 0.352±0.00 | 0.475±0.00 / 0.412±0.00 / 0.478±0.00 | 0.400±0.00 / 0.454±0.01 / 0.399±0.01 | 0.927±0.01 / 0.904±0.01 / 0.897±0.00 | 25.77±0.00 / 29.59±0.01 / 24.52±0.01 | 91.93±0.01 / 94.81±0.00 / 91.35±0.00 | 35.68±0.02 / 43.24±0.02 / 34.65±0.01 | 65.78±0.01 / 81.59±0.01 / 64.53±0.03 | 0.965±0.00 / 0.964±0.01 / 0.964±0.00 | 10.28±0.04 / 10.53±0.01 |
| | + Seq. | 0.336±0.00 / 0.353±0.01 / 0.344±0.00 | 0.454±0.00 / 0.420±0.00 / 0.456±0.00 | 0.390±0.00 / 0.450±0.01 / 0.372±0.01 | 0.859±0.02 / 0.886±0.01 / 0.864±0.02 | 34.65±0.01 / 37.74±0.01 / 36.17±0.01 | 96.43±0.00 / 96.44±0.00 / 96.30±0.00 | 48.88±0.00 / 52.34±0.01 / 49.86±0.03 | 74.36±0.01 / 77.46±0.01 / 74.38±0.01 | 0.962±0.00 / 0.962±0.01 / 0.960±0.00 | 9.84±0.07 / 10.07±0.04 |
| | + κ,α | 0.347±0.00 / 0.373±0.00 / 0.364±0.00 | 0.488±0.00 / 0.462±0.00 / 0.494±0.00 | 0.409±0.02 / 0.455±0.01 / 0.401±0.00 | 0.860±0.01 / 0.900±0.01 / 0.864±0.01 | 37.76±0.01 / 41.48±0.02 / 37.35±0.00 | 96.72±0.00 / 97.29±0.00 / 96.82±0.01 | 51.12±0.02 / 56.20±0.01 / 51.24±0.02 | 78.97±0.01 / 82.70±0.00 / 78.24±0.01 | 0.965±0.00 / 0.963±0.00 / 0.964±0.00 | 8.89±0.04 / 9.65±0.03 |
| | + φ,ψ,ω | 0.359±0.00 / 0.361±0.00 / 0.364±0.01 | 0.485±0.00 / 0.424±0.00 / 0.490±0.00 | 0.396±0.01 / 0.431±0.03 / 0.389±0.01 | 0.861±0.02 / 0.907±0.01 / 0.875±0.02 | 37.90±0.02 / 40.54±0.02 / 39.67±0.01 | 96.06±0.01 / 96.57±0.00 / 96.67±0.00 | 52.46±0.01 / 56.29±0.02 / 53.38±0.02 | 78.01±0.02 / 82.12±0.01 / 78.74±0.01 | 0.964±0.00 / 0.962±0.00 / 0.963±0.00 | |
| | + χ₁₋₄ | 0.350±0.01 / 0.338±0.00 / 0.360±0.01 | 0.482±0.01 / 0.381±0.01 / 0.483±0.00 | 0.406±0.02 / 0.432±0.00 / 0.397±0.02 | 0.881±0.01 / 0.899±0.01 / 0.889±0.01 | 36.30±0.01 / 38.25±0.01 / 37.08±0.02 | 96.23±0.01 / 95.77±0.00 / 96.28±0.01 | 52.23±0.02 / 49.70±0.01 / 51.74±0.02 | 76.98±0.01 / 80.43±0.01 / 75.20±0.07 | 0.963±0.00 / 0.962±0.00 / 0.961±0.00 | |
| GCPNet | Cα | 0.321±0.00 / 0.300±0.01 / 0.322±0.00 | 0.434±0.00 / 0.392±0.01 / 0.450±0.00 | 0.415±0.00 / 0.435±0.01 / 0.430±0.00 | 0.881±0.02 / 0.846±0.04 / 0.846±0.02 | 32.76±0.00 / 36.47±0.01 / 30.82±0.00 | 93.59±0.00 / 95.64±0.00 / 93.01±0.01 | 41.10±0.02 / 49.67±0.00 / 41.25±0.01 | 66.97±0.01 / 76.47±0.00 / 68.32±0.02 | 0.968±0.00 / 0.960±0.00 / 0.967±0.00 | |
| | + Seq. | 0.295±0.02 / 0.319±0.00 / 0.308±0.01 | 0.420±0.01 / 0.399±0.02 / 0.422±0.01 | 0.391±0.00 / 0.423±0.03 / 0.389±0.00 | 0.844±0.01 / 0.824±0.02 / 0.831±0.01 | 36.97±0.01 / 38.51±0.01 / 34.67±0.01 | 95.65±0.00 / 96.05±0.00 / 95.68±0.00 | 47.35±0.01 / 51.85±0.02 / 46.39±0.01 | 73.00±0.02 / 72.18±0.02 / 72.59±0.02 | 0.968±0.00 / 0.961±0.00 / 0.967±0.00 | 8.35±0.08 / 8.92±0.07 |
| | + κ,α | 0.364±0.00 / 0.364±0.02 / 0.371±0.00 | 0.465±0.00 / 0.425±0.01 / 0.468±0.01 | 0.427±0.00 / 0.442±0.01 / 0.422±0.00 | 0.838±0.02 / 0.812±0.01 / 0.829±0.04 | 36.97±0.02 / 38.86±0.02 / 38.42±0.00 | 96.53±0.00 / 96.35±0.00 / 96.55±0.00 | 48.89±0.01 / 51.78±0.00 / 50.77±0.01 | 76.46±0.01 / 76.89±0.01 / 75.46±0.01 | 0.966±0.00 / 0.962±0.00 / 0.967±0.00 | 8.80±0.09 / 9.49±0.18 |
| | + φ,ψ,ω | 0.362±0.01 / 0.345±0.02 / 0.369±0.00 | 0.466±0.01 / 0.431±0.02 / 0.470±0.00 | 0.424±0.00 / 0.436±0.01 / 0.426±0.01 | 0.834±0.01 / 0.815±0.03 / 0.830±0.01 | 38.34±0.01 / 38.86±0.02 / 38.42±0.00 | 95.94±0.00 / 96.25±0.00 / 96.03±0.00 | 49.81±0.01 / 50.96±0.01 / 50.28±0.00 | 75.49±0.01 / 77.01±0.01 / 77.71±0.01 | 0.967±0.00 / 0.962±0.00 / 0.967±0.00 | 7.56±0.11 / 8.60±0.09 |
| | + χ₁₋₄ | 0.329±0.01 / 0.334±0.02 / 0.350±0.00 | 0.456±0.01 / 0.390±0.02 / 0.453±0.02 | 0.410±0.01 / 0.431±0.00 / 0.421±0.00 | 0.855±0.00 / 0.867±0.02 / 0.862±0.01 | 35.32±0.02 / 36.89±0.01 / 37.05±0.01 | 94.80±0.01 / 94.84±0.00 / 95.88±0.00 | 47.06±0.02 / 46.08±0.01 / 47.65±0.01 | 73.00±0.03 / 74.35±0.02 / 71.78±0.04 | 0.968±0.00 / 0.962±0.00 / 0.964±0.00 | |
| TFN | Cα | 0.374±0.00 / 0.371±0.00 / 0.375±0.00 | 0.489±0.00 / 0.447±0.00 / 0.489±0.01 | 0.421±0.00 / 0.452±0.00 / 0.429±0.00 | 0.923±0.01 / 0.906±0.00 / 0.921±0.01 | 25.12±0.00 / 30.71±0.00 / 23.80±0.02 | 91.88±0.00 / 95.70±0.01 / 90.03±0.02 | 34.26±0.01 / 46.34±0.01 / 31.73±0.01 | 69.22±0.02 / 81.22±0.01 / 67.67±0.01 | 0.967±0.00 / 0.960±0.00 / 0.966±0.00 | |
| | + Seq. | 0.332±0.00 / 0.355±0.00 / 0.341±0.00 | 0.429±0.01 / 0.427±0.00 / 0.431±0.00 | 0.396±0.00 / 0.435±0.00 / 0.402±0.00 | 0.867±0.01 / 0.881±0.01 / 0.871±0.00 | 30.11±0.01 / 33.50±0.01 / 31.78±0.01 | 93.89±0.00 / 94.70±0.00 / 94.03±0.00 | 41.58±0.00 / 45.22±0.01 / 41.51±0.01 | 73.69±0.01 / 71.27±0.02 / 69.58±0.04 | 0.961±0.00 / 0.960±0.00 / 0.964±0.00 | 10.34±0.03 / 10.84±0.04 |
| | + κ,α | 0.380±0.00 / 0.365±0.00 | 0.468±0.00 / 0.465±0.00 / 0.472±0.00 | 0.408±0.00 / 0.438±0.00 / 0.418±0.00 | 0.865±0.02 / 0.904±0.01 / 0.853±0.01 | 32.68±0.01 / 36.65±0.01 / 33.63±0.02 | 95.53±0.01 / 96.03±0.00 / 95.85±0.00 | 46.73±0.01 / 49.79±0.00 / 46.79±0.01 | 75.39±0.02 / 78.54±0.01 / 75.67±0.02 | 0.965±0.00 / 0.961±0.00 / 0.964±0.00 | 10.02±0.05 / 10.46±0.11 |
| | + φ,ψ,ω | 0.376±0.00 / 0.367±0.00 | 0.470±0.01 / 0.444±0.01 / 0.476±0.00 | 0.405±0.00 / 0.449±0.00 / 0.417±0.01 | 0.861±0.03 / 0.901±0.01 / 0.875±0.00 | 32.99±0.01 / 36.20±0.01 / 34.48±0.02 | 95.41±0.01 / 95.64±0.00 / 96.09±0.00 | 47.23±0.01 / 52.98±0.01 / 48.58±0.02 | 80.84±0.01 / 76.17±0.02 / 76.12±0.02 | 0.967±0.00 / 0.963±0.00 / 0.966±0.00 | 8.73±0.02 / 9.73±0.01 |
| | + χ₁₋₄ | 0.351±0.00 / 0.367±0.00 | 0.461±0.01 / 0.416±0.00 / 0.471±0.00 | 0.407±0.01 / 0.429±0.00 / 0.418±0.00 | 0.860±0.00 / 0.902±0.01 / 0.862±0.01 | 33.18±0.01 / 32.61±0.01 / 32.67±0.01 | 94.95±0.00 / 94.52±0.01 / 95.26±0.00 | 47.09±0.02 / 46.08±0.01 / 47.65±0.01 | 78.22±0.01 / 74.58±0.01 / 72.68±0.01 | 0.965±0.00 / 0.962±0.00 / 0.964±0.00 | |
| MACE | Cα | —— / 0.350±0.00 | —— / 0.457±0.00 | —— / 0.411±0.01 | 0.913±0.01 / 0.908±0.01 / 0.914±0.01 | 28.02±0.00 / 29.76±0.02 / 28.57±0.01 | 90.18±0.01 / 93.21±0.02 / 89.28±0.01 | 37.77±0.02 / 43.61±0.01 / 37.25±0.01 | —— / 62.38±0.01 | 0.964±0.00 / 0.960±0.00 / 0.964±0.00 | |
| | + Seq. | —— / 0.317±0.01 | —— / 0.434±0.00 | —— / 0.394±0.00 | 0.821±0.02 / 0.891±0.02 / 0.807±0.03 | 31.26±0.01 / 33.50±0.01 / 31.78±0.01 | 93.88±0.02 / 94.90±0.00 / 92.87±0.01 | 41.14±0.03 / 43.19±0.01 / 41.43±0.02 | —— / 69.32±0.01 | 0.964±0.00 / 0.960±0.00 / 0.963±0.00 | 9.95±0.10 / 10.3±0.04 |
| | + κ,α | —— / 0.340±0.01 | —— / 0.453±0.01 | —— / 0.393±0.00 | 0.850±0.03 / 0.904±0.01 / 0.812±0.02 | 32.97±0.03 / 33.24±0.01 / 31.26±0.02 | 93.26±0.01 / 94.49±0.01 / 93.82±0.01 | 42.44±0.01 / 45.24±0.00 / 43.95±0.01 | —— / 74.34±0.02 | 0.965±0.00 / 0.960±0.00 / 0.963±0.00 | 10.30±0.02 / 10.60±0.02 |
| | + φ,ψ,ω | —— / 0.306±0.05 | —— / 0.457±0.00 | —— / 0.404±0.00 | 0.822±0.01 / 0.893±0.02 / 0.830±0.02 | 35.64±0.02 / 34.93±0.03 / 33.56±0.02 | 95.94±0.01 / 95.35±0.01 / 95.46±0.01 | 48.16±0.02 / 48.27±0.02 / 46.57±0.01 | —— / 76.34±0.01 | 0.965±0.00 / 0.960±0.00 / 0.964±0.00 | 8.94±0.03 / 9.75±0.07 |
| | + χ₁₋₄ | —— / 0.308±0.05 | —— / 0.457±0.01 | —— / 0.397±0.01 | 0.822±0.02 / 0.918±0.00 / 0.820±0.02 | 33.65±0.03 / 32.47±0.01 / 35.68±0.03 | 95.39±0.01 / 93.93±0.00 / 95.61±0.00 | 47.16±0.02 / 44.30±0.01 / 47.40±0.02 | —— / 76.10±0.02 | 0.965±0.00 / 0.960±0.00 / 0.964±0.00 | |

Table 3: **Validation performance for pretraining tasks on AlphaFoldDB.** Metrics: Inverse Folding: perplexity; pLDDT, Structure Denoising, Torsional Denoising: RMSE; Seq. Denoising: Accuracy. Best (second-best) results are **bolded** (underlined). **Key takeaway:** Incorporating  backbone structural features  (i.e., adding torsion angles $\phi, \psi, \omega$ ), in general, improves pretraining performance compared to using only  virtual angles  along the sequence.

| Method | Task | | | | |
|---|---|---|---|---|---|
| | Inverse Folding (↓) | pLDDT Pred. (↓) | Structure Denoising (↓) | Seq. Denoising (↑) | Torsional Denoising (↓) |
| **Cα + κ, α** | | | | | |
| SchNet | 7.791 | 0.2397 | 0.0704 | 36.81 | 0.0586 |
| GearNet-Edge | 6.596 | **0.2326** | 0.0672 | 43.76 | 0.0615 |
| EGNN | 6.016 | 0.2406 | 0.0700 | 40.51 | 0.0586 |
| GCPNet | 6.243 | 0.2395 | 0.0679 | 44.81 | 0.0562 |
| **Cα + κ, α, φ, ψ, ω** | | | | | |
| SchNet | 5.562 | 0.2388 | 0.0603 | 45.61 | 0.0489 |
| GearNet-Edge | 5.324 | 0.2402 | 0.0562 | 50.15 | 0.0538 |
| EGNN | 5.962 | 0.2403 | 0.0593 | 53.80 | 0.0487 |
| GCPNet | **3.839** | 0.2399 | **0.0561** | **59.54** | **0.0443** |

## 4.3 Pre-training and Greater Structural Detail Benefit Downstream Tasks

Following the observation that more fine-grained input representations improve pretraining performance, Table 4 explores finetuning on downstream tasks:

- **Whether these lessons from pretraining translate to downstream tasks?** Equivariant GNNs outperform invariant GNNs in the majority of cases and generally benefit the most from pretraining on structure-based tasks, particularly when provided with greater structural detail in input features.

- **Which combination of parameters performs best on downstream tasks?** Overall, providing a greater amount of structural detail compared to a strict Cα atom representation benefits downstream performance for *both* invariant and equivariant models. Notably, structure denoising generally improves downstream performance for *both* types of models.

Table 4: **Pretrained model benchmark results.** Results for each model and featurisation pair are given as:  no pretraining  /  sequence denoising  /  structure denoising , except for inverse folding on CATH, which is pretrained with  inverse folding  on AFDB. **Key takeaway:** The equivariant GCPNet model benefits most from pretraining and maximum structural detail.

| Method | Features | GO-BP (↑) | GO-MF (↑) | GO-CC (↑) | Fold (↑) | | | Reaction (↑) | Inverse Folding (↓) |
|---|---|---|---|---|---|---|---|---|---|
| | | | | | Fold | Family | Superfamily | | |
| GearNet | $C\alpha + \kappa, \alpha$ | 0.393 / 0.342 / 0.376 | 0.476 / 0.481 / **0.490** | 0.436 / 0.446 / **0.457** | 32.79 / 29.33 / 34.00 | 95.35 / 91.72 / 96.57 | 47.56 / 43.40 / 49.63 | 77.45 / 78.64 / 79.37 | 12.35 / 7.84 |
| | $C\alpha + \kappa, \alpha, \phi, \psi, \omega$ | **0.397** / 0.378 / 0.392 | 0.480 / 0.479 / 0.497 | 0.441 / 0.445 / 0.457 | 33.75 / 31.20 / 36.47 | 94.35 / 94.58 / 94.02 | 46.60 / 45.83 / 48.23 | 76.61 / 77.22 / **80.31** | 11.61 / 7.29 |
| GCPNet | $C\alpha + \kappa, \alpha$ | 0.364 / 0.358 / 0.336 | 0.465 / 0.484 / 0.451 | 0.427 / 0.414 / 0.404 | 36.97 / 37.57 / 41.81 | 95.65 / **96.82** / 96.51 | 47.35 / 53.45 / 54.95 | 76.46 / 78.89 / 78.84 | 8.80 / 7.37 |
| | $C\alpha + \kappa, \alpha, \phi, \psi, \omega$ | 0.362 / 0.348 / 0.334 | 0.466 / 0.501 / **0.502** | 0.424 / 0.409 / 0.404 | 38.34 / 41.14 / **42.81** | 95.94 / 96.09 / 96.61 | 49.81 / 52.60 / 57.17 | 75.49 / 77.97 / 79.18 | 7.56 / **6.55** |

## 5 Conclusions

This work focuses on building a comprehensive and multi-task benchmark for protein structure representation learning. *ProteinWorkshop* provides a unified implementation of large pretraining corpora, featurisation schemes, Geometric GNN models and benchmarking tasks to evaluate the effectiveness of protein structure encoding methods. Key findings include that structural pretraining, as well as auxiliary sequence and structure denoising tasks, improve performance on a wide range of downstream tasks and that incorporating more structural detail in featurisation improves performance. Our benchmark is flexible for including new tasks and datasets and is open to the wider research community.

**Availability.** The *ProteinWorkshop* codebase is available under a permissive MIT License at github.com/a-r-j/ProteinWorkshop and accompanying documentation, preprocessed datasets and pretrained model weights are hosted publicly at proteins.sh. Preprocessed datasets and pretrained model weights are deposited on Zenodo at the following URLs, respectively: zenodo.org/record/8282470 and zenodo.org/record/8287754.

## ACKNOWLEDGEMENTS

We acknowledge that this work was supported by a variety of institutions. ARJ was funded by a Biotechnology and Biological Sciences Research Council (BBSRC) DTP studentship (BB/M011194/1). AM and JC were supported by a U.S. NSF grant (DBI2308699) and two U.S. NIH grants (R01GM093123 and R01GM146340). CKJ was supported by the A*STAR Singapore National Science Scholarship (PhD). This work was performed using resources provided by the Cambridge Service for Data Driven Discovery (CSD3) operated by the University of Cambridge Research Computing Service (`www.csd3.cam.ac.uk`), provided by Dell EMC and Intel using Tier-2 funding from the Engineering and Physical Sciences Research Council (capital grant EP/T022159/1), and DiRAC funding from the Science and Technology Facilities Council (`www.dirac.ac.uk`). Additionally, this work was performed using high performance computing infrastructure provided by Research Support Services at the University of Missouri-Columbia (DOI: 10.32469/10355/97710). We thank Martin Steinegger and Do-Yoon Kim for allowing us to use the illustrations in Figure 1.

## BROADER IMPACTS

Our benchmark unifies protein representation learning tasks, large-scale pre-training datasets, featurisation schemes, and models. The wide range of tasks studied in our benchmark can enable us to develop insight into effective pre-training strategies, and whether pre-trained protein structural representations can have material impact in real-world computational biology and drug discovery. It is not lost on us that these models can play a role in developing harmful chemical matter in the hands of a bad actor. Additionally, training very large models can contribute to climate change. However, we hope that developing highly effective structural representations will have broad, positive implications across biology and medicine that significantly outweigh the potential for misuse.

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

APPENDICES

# A ILLUSTRATED RESULTS

Following You et al. (2020), in this section, we provide an alternative means of interpreting the main results in Table 2 of the main text. Across Figures 2-10, we illustrate the test metric stability of each encoder, featurization scheme, and auxiliary task across each dataset included in Table 2 via a ranking analysis. For each configuration, we rank design choices by their performance, deeming performance to be tied when the difference $\epsilon < 0.02$ ($\epsilon < 0.001$ for PPI site prediction). We collect rankings over all configurations and report the mean ranking and their smoothed distribution for each task with bar and violin plots, respectively (lower is better). An average ranking score of 1 indicates the design choice invariably results in the highest performance over other possibilities in its category. The smoothed distributions provide further insight into the broader behaviour of a design choice.

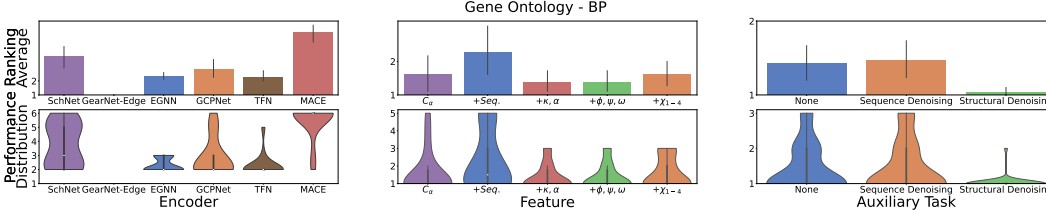

Figure 2: Ranking analysis of Gene Ontology-Biological Process (GO-BP) test performance across different encoders, feature sets, and auxiliary tasks.

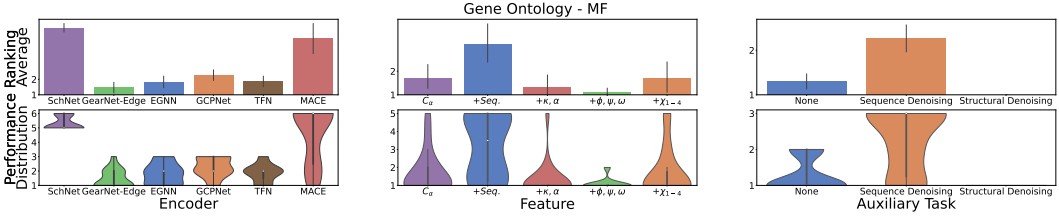

Figure 3: Ranking analysis of Gene Ontology-Molecular Function (GO-MF) test performance across different encoders, feature sets, and auxiliary tasks.

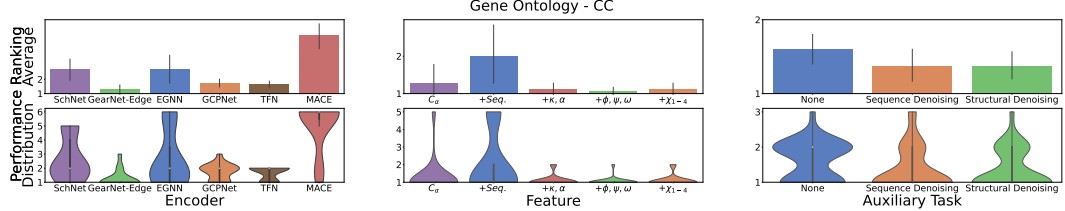

Figure 4: Ranking analysis of Gene Ontology-Cellular Component (GO-CC) test performance across different encoders, feature sets, and auxiliary tasks.

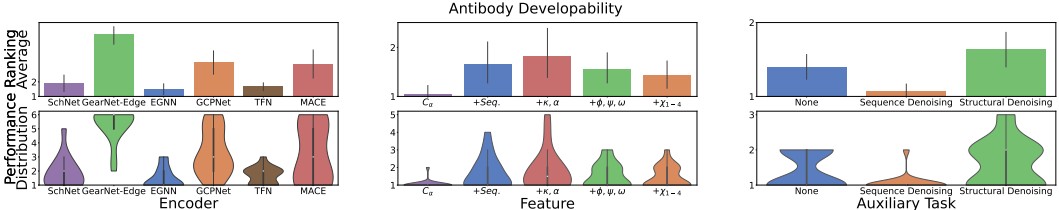

Figure 5: Ranking analysis of Antibody Developability test performance across different encoders, feature sets, and auxiliary tasks.

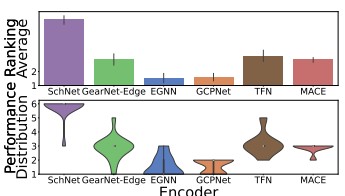
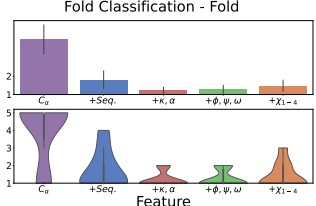
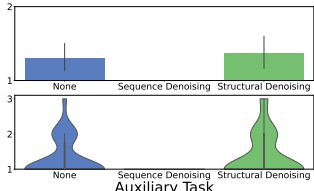

Figure 6: Ranking analysis of Fold Classification-Fold test performance across different encoders, feature sets, and auxiliary tasks.

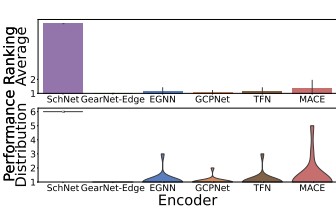
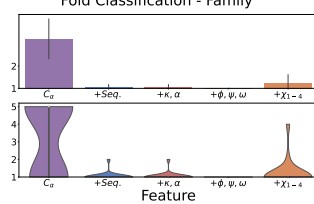
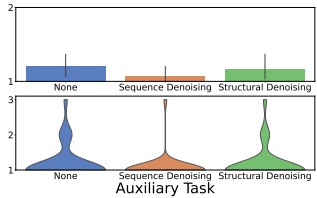

Figure 7: Ranking analysis of Fold Classification-Family test performance across different encoders, feature sets, and auxiliary tasks.

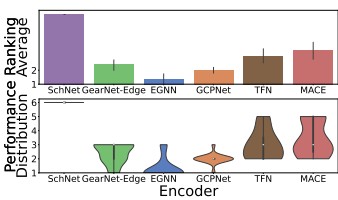
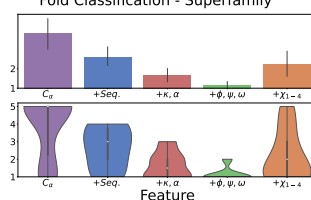
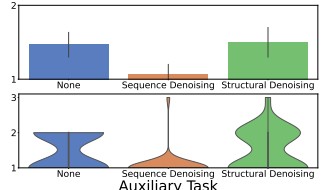

Figure 8: Ranking analysis of Fold Classification-Superfamily test performance across different encoders, feature sets, and auxiliary tasks.

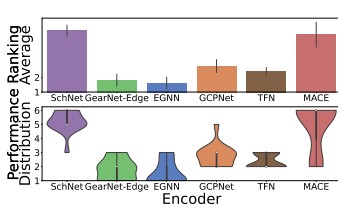
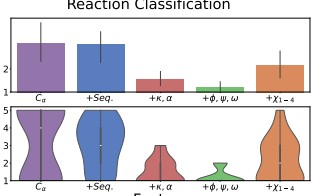
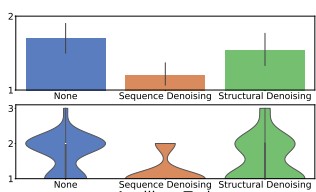

Figure 9: Ranking analysis of Reaction Classification test performance across different encoders, feature sets, and auxiliary tasks.

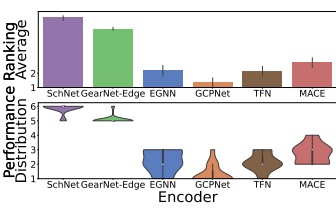
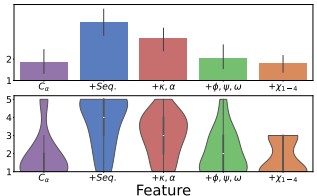
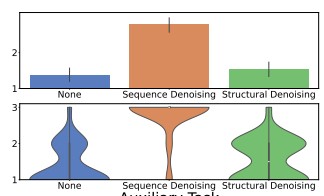

Figure 10: Ranking analysis of Protein-Protein Interaction (PPI) Site Prediction test performance across different encoders, feature sets, and auxiliary tasks.

## B  RELATED WORK

**Protein Structure Representation Learning.**  Several structure-based encoders for proteins have been designed to extract information from different levels of granularity, such as residue-level, atom-level and surfaces. Previous works have aimed to encode protein structural priors directly within architectures to model proteins hierarchically (Somnath et al., 2021; Hermosilla et al., 2020), as computationally efficient point clouds (Gainza et al., 2020; Sverrisson et al., 2021), or as geometric graphs (Jing et al., 2020; Jin et al., 2021; Morehead & Cheng, 2024; Wang et al., 2023; Zhang et al., 2023b; Mahmud et al., 2023) for tasks such as protein function prediction (Gligorijević et al., 2021), protein model quality assessment (Eismann et al., 2020; Chen et al., 2023; Morehead et al., 2024), and protein interaction region prediction (Dai & Bailey-Kellogg, 2021; Morehead et al., 2023a).

**Protein Benchmarks.**  Several benchmarks have been proposed for evaluating the efficacy of learnt protein *sequence* representations. However, *structure-based* benchmarks are comparatively unaddressed. Rao et al. (2019) developed TAPE (Tasks Assessing Protein Embeddings), providing a large pretraining corpus of protein sequences curated from Pfam (El-Gebali et al., 2018), as well as a collection of five supervised benchmark tasks assessing the ability of protein language models to predict structural qualities (contact prediction and secondary structure prediction), and functional properties (fluorescence and stability prediction). Xu et al. (2022) proposed PEER (Protein Sequence Understanding), focussing on multitask evaluation of protein sequence models. Therapeutic Data Commons (Huang et al., 2021) provide several datasets relevant to therapeutic development, however the few protein structure-derived datasets it contains are cast as sequence-based tasks. Dallago et al. (2021) developed FLIP, a sequence-based benchmark of protein fitness landscapes. ProteinGLUE (Capel et al., 2022) is another sequenced-based benchmark focussing on per-residue tasks.

To our best knowledge, the only protein structure-benchmark to date is ATOM3D (Townshend et al., 2021), which proposes a collection of tasks largely assessing geometric methods at predicting graph-level properties of protein structures. TorchProtein (Zhu et al., 2022) also provides a small collection of global-structural datasets. Most existing benchmarks do not exhaustively evaluate both the local and global representation learning power of proposed methods. As the field develops, we identify a need for a consistent benchmarking framework of diverse tasks to ensure improving results reported in the literature map on to progress in the downstream problems we hope to address. Similar benchmarking efforts for general purpose GNNs have provided experimental rigour to architectural research (Dwivedi et al., 2023).

**Denoising-Based Pre-training and Regularisation.**  Several methods have been developed for pre-training GNNs, predominantly focussing on cases where 3D coordinate information is only implicitly encoded in the graph structures. In this work, we build on work by Godwin et al. (2021) and Zaidi et al. (2023) to investigate whether denoising-based auxillary and pre-training tasks are effective methods for pre-training geometric GNNs operating on protein structures, similar to concurrent works bridging the gap between denoising objectives for geometric neural networks and diffusion generative modeling for biomolecules (Huang et al., 2023; Corso, 2023).

## C  DISCUSSION AND FUTURE WORK

**Completeness of input featurisation.**  The extent to which providing complete information about all atoms and side chain orientations of each residue in a protein is debatable, as the exact coordinates from PDB files are known to contain artifacts from structure determination via crystallography. This was most recently noted by Dauparas et al. (2022) in the context of developing and experimentally validating an inverse folding model. Our benchmarking results provides similar insights – at present, letting models implicitly learn about side chain orientation by using backbone-only featurisation performs better or equally well as explicitly providing complete side chain atomic information across both global and node level tasks.

**On the choice of pre-training tasks.**  We currently focussed on pre-training tasks that roughly fall under the category of denoising, i.e. corrupting information in the input (sequence identity, coordinates) and tasking the model with producing the uncorrupted input. We were particularly interested in self-supervised objectives that were (1) extremely scalable, so as to pre-train on the large-scale AlphaFoldDB of 2.4M structures; and (2) train protein representations at the fine-grained node level, so as to be general-purpose across the downstream tasks considered. We did not benchmark other

tasks from the literature, such as contrastive learning and generative modelling-inspired objectives (Liu et al., 2023b;a; Zhang et al., 2023a). Such tasks are generally (1) computationally heavier and more cumbersome to set up than corruption-type objectives, making them harder to scale up, and (2) only train protein representations for the global/graph level and do not operate at the node level. Naturally, we would like to continue exploring more pre-training strategies as we continue to expand *ProteinWorkshop*.

**Beyond protein-only representation learning.** Recently, Krishna et al. (2023) and DeepMind-Isomorphic (2023) generalised protein structure prediction models to full biological assemblies including proteins, small molecules, nucleic acids, and other ligands. Geometric graphs are a natural choice for representation learning across biomolecular systems: Advances in geometric GNN methodology should, in principle, be adaptable to modelling other biomolecules (Joshi et al., 2023b), their complexes (Morehead et al., 2023b), and quaternary structures.

We currently focus on protein representation learning because (1) large scale datasets for self-supervised learning, as well as well-defined downstream tasks, are readily available and accepted by the community; and (2) we see protein representation learning as a fundamental task, improving upon which should also advance the modelling of proteins in complex with other biomolecules. Comparatively, the scale of data available for biomolecular complexes is smaller and there is less consensus among the community on evaluation (Harris et al., 2023; Buttenschoen et al., 2023).

# D  *ProteinWorkshop* USER MANUAL

## D.1  DEPENDENCIES

The benchmark is developed using `PyTorch` (Paszke et al., 2019), `PyTorch Geometric` (Fey & Lenssen, 2019), `PyTorch Lightning` (Falcon, 2019), and `Graphein` (Jamasb et al., 2022). Experiment configuration is performed using `Hydra` (Yadan, 2019). Certain architectures introduce additional dependencies, such as `TorchDrug` (Zhu et al., 2022) and `e3nn` (Geiger & Smidt, 2022).

## D.2  USAGE

The modular design of our benchmark means it can be readily adapted into different workflows easily. Firstly, the benchmark is `pip`-installable from PyPI and contains several importable modules, such as dataloaders, featurisers and models, that can be imported into new projects. This will aid in standardising the datasets and workflows used in protein representation learning. Secondly, the benchmark serves as an easily extendable template, which users can fork and work directly in, thereby reducing implementation burden. Lastly, we provide a CLI that can be used to quickly run single experiments and hyperparameter sweeps with minimal development time overhead.

## D.3  COMPUTATIONAL RESOURCES

All models are trained on 80GB NVIDIA A100 GPUs. All baseline and finetuning results are performed using a single GPU while pre-training is performed using four GPUs.

## D.4  FEATURISATION SCHEMES

Protein structures are typically represented as graphs, with researchers typically opting to use a coarse-grained $C\alpha$ atoms graph as full atom representations can quickly become computationally intractable due to a large number of nodes. The extent to which coarse-graining schemes are 'complete' representation of the geometry and structure of the protein residue is variable. For instance, backbone-only features ignore the orientations of the side chain atoms in the residue, so models must account for this information implicitly. However, providing complete information about all atoms and side chain orientations is debatable as exact coordinates from PDB files are known to contain crystallography artefacts (Dauparas et al., 2022). Due to the computational burden incurred by operating on full-atom node representations, we focus primarily on $C\alpha$-based graph representations, investigating featurisation strategies to incorporate higher-level structural information. Note that we do provide utilities to enable users to work with backbone and full-atom graphs in the benchmark.

We represent protein structures as geometric graphs, $\mathcal{G} = (\mathcal{V}, \mathcal{E}, \vec{\mathbf{X}}, \mathbf{S}, \vec{\mathbf{V}})$, where $\mathcal{V}$ is a set of nodes, $\mathcal{E}$ is a set of edges, $\vec{\mathbf{X}} \in \mathbb{R}^{|\mathcal{V}| \times 3}$ is a matrix of Cartesian node coordinates, $\mathbf{S} \in \mathbb{R}^{|\mathcal{V}| \times d}$ is a matrix of $d$-dimension scalar node features, and $\vec{\mathbf{V}} \in \mathbb{R}^{|\mathcal{V}| \times d \times 3}$ is a tensor of vector-valued features.

The five scalar featurisation schemes considered in the baselines are provided in Table 5. Pretraining experiments are performed on the featurisation schemes in rows three and four.

Table 5: **Structural featurisation schemes.** Residue type is a one-hot encoding of the amino acid type for each node; positional encoding is a 16-dimensional transformer-like positional encoding (Vaswani et al., 2017); and $\phi, \psi, \omega \in \mathbb{R}^6$ and $\chi_{1-4} \in \mathbb{R}^8$ are backbone dihedral angles and sidechain torsion angles, respectively, embedded on the unit circle. Similarly, $\kappa, \alpha \in \mathbb{R}^4$ are *virtual* torsion and bond angles defined over $C\alpha$ atoms. In our experimental evaluation, we consistently use $k$-NN edge construction with $k = 16$.

| Granularity | $C\alpha$ Features | Backbone | Sidechain |
|---|---|---|---|
| $C\alpha$ | Residue Type | | |
| $C\alpha$ | Residue Type, Positional Encoding | | |
| $C\alpha$ | Residue Type, Positional Encoding, $\kappa, \alpha$ | | |
| $C\alpha$ | Residue Type, Positional Encoding, $\kappa, \alpha$ | $\phi, \psi, \omega$ | |
| $C\alpha$ | Residue Type, Positional Encoding, $\kappa, \alpha$ | $\phi, \psi, \omega$ | $\chi_1, \chi_2, \chi_3, \chi_4$ |

Additionally, vector message passing methods such as GCPNet receive node orientation vectors (i.e., $\boldsymbol{v}_{o^{i-1}}$ and $\boldsymbol{v}_{o^{i+1}}$) and edge directional vectors (i.e., $\boldsymbol{v}_{d^{ij}}$) as vector input features for nodes and edges, respectively. These vector features, prior to being normalised as unit vectors, are defined as:

$$\vec{\boldsymbol{v}}_{o^{i-1}} = \vec{\boldsymbol{x}}_{i-1} - \vec{\boldsymbol{x}}_i, \ \vec{\boldsymbol{v}}_{o^{i+1}} = \vec{\boldsymbol{x}}_{i+1} - \vec{\boldsymbol{x}}_i, \ \text{and} \ \vec{\boldsymbol{v}}_{d^{ij}} = \vec{\boldsymbol{x}}_i - \vec{\boldsymbol{x}}_j. \tag{1}$$

## D.5 PROTEIN STRUCTURE ENCODER ARCHITECTURES

We provide a unified implementation of several rotation invariant and equivariant geometric GNNs, spanning the range of message passing body order and tensor order (Joshi et al., 2023a), including general-purpose as well as protein-specific models. See Duval et al. (2023) for a self-contained introduction to geometric GNNs.

### D.5.1 INVARIANT GNNS

**SchNet (Schütt et al., 2018)** SchNet is one of the most popular and simplest instantiation of E(3) invariant message passing GNNs. SchNet constructs messages through element-wise multiplication of scalar features modulated by a radial filter conditioned on the pairwise distance $\|\vec{\boldsymbol{x}}_{ij}\|$ between two neighbours. Scalar features are update from iteration $t$ to $t + 1$ via:

$$\boldsymbol{s}_i^{(t+1)} := \boldsymbol{s}_i^{(t)} + \sum_{j \in \mathcal{N}_i} f_1\left(\boldsymbol{s}_j^{(t)}, \|\vec{\boldsymbol{x}}_{ij}\|\right) \tag{2}$$

Hyperparameters: number of layers = 6, hidden channels = 512.

**DimeNet++ (Gasteiger et al., 2020)** DimeNet is an E(3) invariant GNN which uses both distances $\|\vec{\boldsymbol{x}}_{ij}\|$ and angles $\vec{\boldsymbol{x}}_{ij} \cdot \vec{\boldsymbol{x}}_{ik}$ to perform message passing among triplets, as follows:

$$\boldsymbol{s}_i^{(t+1)} := \sum_{j \in \mathcal{N}_i} f_1\left(\boldsymbol{s}_i^{(t)}, \boldsymbol{s}_j^{(t)}, \sum_{k \in \mathcal{N}_i \setminus \{j\}} f_2\left(\boldsymbol{s}_j^{(t)}, \boldsymbol{s}_k^{(t)}, \|\vec{\boldsymbol{x}}_{ij}\|, \vec{\boldsymbol{x}}_{ij} \cdot \vec{\boldsymbol{x}}_{ik}\right)\right) \tag{3}$$

Hyperparameters: number of layers = 6, hidden channels = 512.

**GearNet-Edge (Zhang et al., 2023b)** GearNet-Edge is an SE(3) invariant architecture leveraging relational graph convolutional layers and edge message passing. The original GearNet-Edge formulation presented in Zhang et al. (2023b) operates on multirelational protein structure graphs making use of several edge construction schemes ($k$-NN, euclidean distance and sequence distance based). Our

benchmark contains full capabilities for working with multirelational graphs but use a single edge type (i.e. $|\mathcal{R}| = 1$) in our experiments to enable more direct architectural comparisons.

The relational graph convolutional layer is defined for relation type $r$ as:

$$s_i^{(t+1)} := s_i^{(t)} + \sigma \left( \text{BN} \left( \sum_{r \in \mathcal{R}} \mathbf{W_r} \sum_{j \in \mathcal{N}_r(i)} s_j^{(t)} ) \right) \right) \tag{4}$$

The edge message passing layer is defined for relation type $r$ as:

$$m_{(i,j,r_1)}^{(t+1)} := \sigma \left( \text{BN} \left( \sum_{r \in |R|'} \mathbf{W}_r' \sum_{(w,k,r_2) \in \mathcal{N}_r'((i,j,r_1))} m_{(w,k,r_2)}^{(t)} \right) \right) \tag{5}$$

$$s_i^{(t+1)} := \sigma \left( \text{BN} \left( \sum_{r \in |R|} \mathbf{W}_r \sum_{j \in \mathcal{N}_r(i)} \left( s_j^{(t)} + \text{FC}(m_{(i,j,r)}^{(t+1)}) \right) \right) \right), \tag{6}$$

where $\text{FC}(\cdot)$ denotes a linear transformation upon the message function.

Hyperparameters: number of layers = 6, hidden channels = 512.

### D.5.2 EQUIVARIANT GNNS IN CARTESIAN COORDINATES

**EGNN (Satorras et al., 2021)** We consider E(3) equivariant GNN layers proposed by Satorras et al. (2021) which updates both scalar features $s_i$ as well as node coordinates $\vec{x}_i$, as follows:

$$s_i^{(t+1)} := f_2 \left( s_i^{(t)}, \sum_{j \in \mathcal{N}_i} f_1 \left( s_i^{(t)}, s_j^{(t)}, \|\vec{x}_{ij}^{(t)}\| \right) \right) \tag{7}$$

$$\vec{x}_i^{(t+1)} := \vec{x}_i^{(t)} + \sum_{j \in \mathcal{N}_i} \vec{x}_{ij}^{(t)} \odot f_3 \left( s_i^{(t)}, s_j^{(t)}, \|\vec{x}_{ij}^{(t)}\| \right) \tag{8}$$

Hyperparameters: number of layers = 6, hidden channels = 512.

**GCPNet (Morehead & Cheng, 2024)** GCPNet is an SE(3) equivariant architecture that jointly learns scalar and vector-valued features from geometric protein structure inputs and, through the use of geometry-complete frame embeddings, sensitises its predictions to account for potential changes induced by the effects of molecular chirality on protein structure. In contrast to the original GCPNet formulation presented in Morehead & Cheng (2024), the implementation we provide in the benchmark incorporates the architectural enhancements proposed in Morehead & Cheng (2023) which include the addition of a scalar message attention gate (i.e., $f_a(\cdot)$) and a simplified structure for the model's geometric graph convolution layers (i.e., $f_n(\cdot)$). With geometry-complete graph convolution in mind, for node $i$ and layer $t$, scalar edge features $s_{e^{ij}}^{(t)}$ and vector edge features $v_{e^{ij}}^{(t)}$ are used along with scalar node features $s_{n^i}^{(t)}$ and vector node features $v_{n^i}^{(t)}$ to update each node feature type as:

$$(s_{m^{ij}}^{(t+1)}, v_{m^{ij}}^{(t+1)}) := f_e^{(t+1)} \left( (s_{n^i}^{(t)}, v_{n^i}^{(t)}), (s_{n^j}^{(t)}, v_{n^j}^{(t)}), (f_a^{(t+1)}(s_{e^{ij}}^{(t)}), v_{e^{ij}}^{(t)}), \mathcal{F}_{ij} \right) \tag{9}$$

$$(s_{n^i}^{(t+1)}, v_{n^i}^{(t+1)}) := f_n^{(t+1)} \left( (s_{n^i}^{(t)}, v_{n^i}^{(t)}), \sum_{j \in \mathcal{N}(i)} (s_{m^{ij}}^{(t+1)}, v_{m^{ij}}^{(t+1)}) \right), \tag{10}$$

where the geometry-complete and chirality-sensitive local frames for node $i$ (i.e., its edges) are defined as $\mathcal{F}_{ij} = (a_{ij}, b_{ij}, c_{ij})$, with $a_{ij} = \frac{x_i - x_j}{\|x_i - x_j\|}$, $b_{ij} = \frac{x_i \times x_j}{\|x_i \times x_j\|}$, and $c_{ij} = a_{ij} \times b_{ij}$, respectively.

Hyperparameters: number of layers = 6, hidden scalar channels = 128, hidden vector channels = 16.

### D.5.3 EQUIVARIANT GNNS IN SPHERICAL COORDINATES

**Tensor Field Network (Thomas et al., 2018)**  Tensor Field Networks are E(3) or SE(3) equivariant GNNs that have been successfully used in protein structure prediction (Baek et al., 2021) and protein-ligand docking (Corso et al., 2023). These models use higher order spherical tensors $\tilde{\boldsymbol{h}}_{i,l} \in \mathbb{R}^{2l+1 \times f}$ as node features, starting from order $l = 0$ up to arbitrary $l = L$. The first two orders correspond to scalar features $\boldsymbol{s}_i$ and vector features $\vec{\boldsymbol{v}}_i$, respectively. The higher order tensors $\tilde{\boldsymbol{h}}_i$ are updated via tensor products $\otimes$ of neighbourhood features $\tilde{\boldsymbol{h}}_j$ for all $j \in \mathcal{N}_i$ with the higher order spherical harmonic representations $Y$ of the relative displacement $\frac{\vec{\boldsymbol{x}}_{ij}}{\|\vec{\boldsymbol{x}}_{ij}\|} = \hat{\boldsymbol{x}}_{ij}$:

$$\tilde{\boldsymbol{h}}_i^{(t+1)} := \tilde{\boldsymbol{h}}_i^{(t)} + \sum_{j \in \mathcal{N}_i} Y\left(\hat{\boldsymbol{x}}_{ij}\right) \otimes_{\boldsymbol{w}} \tilde{\boldsymbol{h}}_j^{(t)}, \tag{11}$$

where the weights $\boldsymbol{w}$ of the tensor product are computed via a learnt radial basis function of the relative distance, *i.e.* $\boldsymbol{w} = f\left(\|\vec{\boldsymbol{x}}_{ij}\|\right)$.

Hyperparameters: choice of symmetry = SO(3), number of layers = 4, spherical harmonic tensor order = 2, hidden irreps per tensor type = `64x0e + 64x0o + 8x1e + 8x1o + 4x2e + 4x2o`. We were particularly interested in benchmarking the impact of higher order tensors and SO(3) equivariance.

**MACE**  MACE (Batatia et al., 2022) is a higher order E(3) or SE(3) equivariant GNN originally developed for molecular dynamics simulations. MACE provides an efficient approach to computing high body order equivariant features in the Tensor Field Network framework via Atomic Cluster Expansion: They first aggregate neighbourhood features analogous to equation 11 (the $A$ functions in Batatia et al. (2022) (eq. 9)) and then take $k - 1$ repeated self-tensor products of these neighbourhood features. In our formalism, this corresponds to:

$$\tilde{\boldsymbol{h}}_i^{(t+1)} := \underbrace{\tilde{\boldsymbol{h}}_i^{(t+1)} \otimes_{\boldsymbol{w}} \cdots \otimes_{\boldsymbol{w}} \tilde{\boldsymbol{h}}_i^{(t+1)}}_{k-1 \text{ times}}, \tag{12}$$

Hyperparameters: choice of symmetry = O(3), number of layers = 2, spherical harmonic tensor order = 2, hidden irreps per tensor type = `32x0e + 32x1o + 32x2e`, body order = 4. Note that the number of channels for all tensor types must be the same for MACE, which is restrictive for scaling the depth and number of parameters.

### D.6 PRETRAINING DATASETS

The benchmark contains several large corpora of both experimental and predicted structural data that can be used for pretraining or inference. We provide utilities for configuring supervised tasks and splits directly from the PDB (Berman, 2000). Additionally, we build storage-efficient dataloaders for large pretraining corpora of predicted structures including the AlphaFoldDB and ESM Atlas. We believe our codebase will considerably reduce the barrier to entry for pretraining and working with large structure-based datasets.

### D.6.1 EXPERIMENTAL STRUCTURES

**PDB.** We provide utilities for curating datasets directly from the Protein Data Bank (Berman, 2000). In addition to using the collection in its entirety, users can define filters to subset and split the data using a combination of structural similarity, sequence similarity or temporal strategies. Structures can be filtered by length, number of chains, resolution, deposition date, presence/absence of particular ligands and structure determination method. The benchmark supports working with PDB structures in both `.pdb` and `.mmtf` format (Bradley et al., 2017), which significantly reduces the requirements for data storage.

**CATH.** We provide the dataset derived from CATH 4.2 40% (Knudsen & Wiuf, 2010) non-redundant chains developed by Ingraham et al. (2019) as an additional, smaller, pretraining dataset. These data are split based on random assignment of the CATH topology classifications based on an 80/10/10 split.

**ASTRAL.** ASTRAL (Brenner, 2000) provides compendia of protein *domain* structures, regions of proteins that can maintain their structure and function independently of the rest of the protein. Domains typically exhibit highly-specific functions and can be considered structural building blocks of proteins.

### D.6.2 PREDICTED STRUCTURES

We provide ready-to-go dataloaders for several large-scale collections of predicted structures derived from both AlphaFold2 (Jumper et al., 2021) and ESMFold (Lin et al., 2022). This is facilitated by FoldComp (Kim et al., 2023), a (minimally) lossy compression scheme for predicted protein structures. FoldComp stores protein structures as a collection of discretised dihedral and bond angles which can be used to reconstruct the whole structure using fixed bond lengths and canonical amino acid geometry. FoldComp achieves a disk-space reduction of almost an order of magnitude, describing a residue with only 13 bytes – down from 97 bytes per-residue in a traditional uncompressed format. Whilst lossy, this procedure results in 0.08 Å and 0.14 Å RMSD for backbone and all-atom reconstruction, making it highly suitable for pretraining tasks which use input representations complete up to the backbone. Furthermore, this lightweight format enables the dataloaders in the benchmark to read structures *directly from disk* with no pre-processing or caching required.

**AlphaFoldDB Representative Structures.** This dataset contains 2.27 million representative structures, identified through large-scale structural-similarity-based clustering of the 214 million structures contained in the AlphaFold Database (Varadi et al., 2021) using FoldSeek (van Kempen et al., 2023). We additionally provide a subset of this collection — the so-called dark proteome — corresponding to the 31% of the representative structures that lack annotations.

**ESM Atlas, ESM High Quality.** These datasets are compressed collections of predicted structures produced by ESMFold. ESM Atlas is the full collection of all 772m predicted structures for the MGnify 2023 release (Richardson et al., 2022). ESM High Quality is a curated subset of high confidence (mean pLDDT) structures from the collection.

### D.7 PRETRAINING TASKS

The benchmark contains a comprehensive suite of pretraining tasks. Broadly, these tasks can be categorised into: masked-attribute prediction, denoising-based and contrastive learning-based tasks. In most cases, these tasks can be used as both a pretraining objective or as auxiliary tasks in a downstream supervised task.

**Sequence Denoising.** The benchmark contains two variations based on two sequence corruption processes $C(\tilde{\mathcal{S}}|\mathcal{S}, \nu)$ that receive an amino acid sequence $\mathcal{S} \in [0, 1]^{|\mathcal{V}| \times 23}$ and return a sequence $\mathcal{S} \in [0, 1]^{|\mathcal{V}| \times 23}$ with fraction $\nu$ of its positions corrupted. The first scheme is based on mutating a fraction of the residues to a random amino acid and tasking the model with recovering the uncorrupted sequence. The second is a masked residue prediction task, where a fraction of the residues are altered to a mask value and the model is tasked to recover the uncorrupted sequence.

**Structure Denoising.** We provide two structure-based denoising tasks: coordinate denoising and torsional denoising. In the coordinate denoising task, noise is sampled from a normal or uniform distribution and scaled by noise factor, $\nu \in \mathbb{R}$, and applied to each of the atom coordinates in the structure to ensure structural features, such as backbone or sidechain torsion angles, are also corrupted. The model is then tasked with predicting either the per-node noise or the original uncorrupted coordinates. For the torsional denoising variant, the noise is applied to the backbone torsion angles and Cartesian coordinates are recomputed using pNeRF (AlQuraishi, 2019) and the uncorrupted bond lengths and angles prior to feature computation. Similarly to the coordinate denoising task, the model is then tasked with predicting either the per-residue angular noise or the original dihedral angles.

**Sequence-Structure Co-Denoising.** This is a multitask formulation of the previously described structure and sequence denoising tasks, with separate output heads for denoising each modality.

**Masked Attribute Prediction Tasks** We use inverse folding (Section 2.3.1) as a pretraining task. The benchmark additionally incorporates the distance, angle and dihedral angle masked-attribute

prediction tasks proposed in Zhang et al. (2023b) as well as a backbone dihedral angle prediction task.

**pLDDT Prediction.** Protein structure prediction models typically provide per-residue pLDDT (predicted Local Distance Difference Test) scores as local confidence measures in the quality of the prediction shown to correlate well with disordered regions (Wilson et al., 2022). We formulate a self-supervised node-level regression task on predicted structures, somewhat analogous to structure quality assessment (QA), where the model is tasked with predicting the scaled per-residue pLDDT $y \in [0, 1]$ values.

## D.8 DOWNSTREAM TASKS

We curate several structure-based and sequence-based datasets from the literature and existing benchmarks[†]. The tasks are selected to evaluate not only the *global* structure representational power of each method, but also to evaluate the ability of each method to learn informative *local* representations for residue-level prediction and annotation tasks.

The raw structures are, where possible and accounting for obsolescence, retrieved directly from the PDB (or another structural source) as several processed datasets used by the community discard full atomic coordinates in favour of retaining only $C_\alpha$ positions making them unsuitable for in-depth experimentation. This provides an entry point for users to apply a custom sequence of pre-processing steps, such as deprotonation or fixing missing regions which are common in experimental data.

The following downstream tasks are available in our benchmark. Detailed documentation including composition, splitting details, metrics, and data sheets (Gebru et al., 2021) are available in Appendix E

**Node-level Tasks**

- CATH Inverse folding.
- PPI Site Prediction.
- Metal Binding Site Prediction.
- Post-Translational Modification Site Prediction.

**Graph-level Tasks**

- Fold Prediction.
- Gene Ontology Prediction.
- Reaction Class Prediction.
- Antibody Developability Prediction.

## D.9 SE(3) EQUIVARIANT NOISE PREDICTOR

Similar to Zhang et al. (2023c), for structure-based denoising tasks we use an SE(3) equivariant noise predictor network to predict per-residue perturbations from SE(3) invariant scalar embeddings and corrupted atomic coordinates $\tilde{\boldsymbol{X}}$. Each edge $\boldsymbol{e}_{ij}$ is featurised by concatenating the two adjoining scalar node representations $\tilde{\boldsymbol{s}}_i, \tilde{\boldsymbol{s}}_j$ and the euclidean distance between them $\|\tilde{\boldsymbol{x}}_i - \tilde{\boldsymbol{x}}_j\|_2$. We use a two-layer MLP to produce a score $\boldsymbol{m}_{ij}$:

$$\boldsymbol{m}_{ij} = \mathrm{MLP}(\tilde{\boldsymbol{s}}_i, \tilde{\boldsymbol{s}}_j, \mathrm{MLP}(\|\tilde{\boldsymbol{x}}_i - \tilde{\boldsymbol{x}}_j\|_2)). \quad (13)$$

Subsequently, Equation 13 is used to aggregate normalised directional edge vectors over the neighbourhood $\mathcal{N}_i$ of each node:

$$\boldsymbol{\epsilon}_\theta(\tilde{\mathcal{G}}) = \sum_{j \in \mathcal{N}_i} \boldsymbol{m}_{ij} \cdot \frac{\tilde{\boldsymbol{x}}_i - \tilde{\boldsymbol{x}}_j}{\|\tilde{\boldsymbol{x}}_i - \tilde{\boldsymbol{x}}_j\|_2} \quad (14)$$

[†]To retain focus on *protein* representation learning, we deliberately exclude commonly-used tasks based on protein-small molecule interactions as it is hard to disentangle the effect of the small molecule representation and the potential for bias (Boyles et al., 2019)

Table 6: **Overview of supervised tasks and datasets.**

| | Task | Dataset Origin | Structures | # Train | # Validation | # Test |
|---|---|---|---|---|---|---|
| Node-level | Inverse Folding | Ingraham et al. (Ingraham et al., 2019) | Experimental | 3.9 M | 105 K | 180 K |
| | PPI Site Prediction | Gainza et al. (2020) | Experimental | 478 K | 53 K | 117 K |
| | Metal Binding Site Prediction | | Experimental | 1.1 M | 13.7 K | 29.8 K |
| | Post-Trans. Modification Site Prediction | Yan et al. (2023) | Predicted | 44 K | 2.4 K | 2.5 K |
| Graph-level | Fold Prediction | Hou et al. (Hou et al., 2017) | Experimental | 12.3 K | 0.7 K | 1.3/0.7/1.3 K |
| | Gene Ontology Prediction | Gligorijević et al. (2021) | Experimental | 27.5 K | 3.1 K | 3.0 K |
| | Reaction Class Prediction | Hermosilla et al. (2020) | Experimental | 29.2 K | 2.6 K | 5.6 K |
| | Antibody Developability Prediction | Huang et al. (Huang et al., 2021) | Experimental | 1.7 K | 0.24 K | 0.48 K |

## D.10 HYPERPARAMETER SELECTION

Given the large number of models and featurisation schemes, we try our best to do a consistent and fair hyperparameter search. We fix a consistently high number of layers and large hidden dimension across models, as we wanted to focus on scaling model size and dataset size via pre-training. We use the Fold Classification task to select the best learning rate and dropout per model and featurisation scheme for downstream tasks. While our best performing models sometimes do not outperform the best reported results in the literature, we have obtained these results in a consistent experimental setup. Our goal was to demonstrate the utility of our benchmarking framework and uncover the impact of architectural considerations such as featurisation schemes, geometric GNN models, and pre-training/auxiliary tasks under fair and rigorous experimental settings.

**Protein Structure Encoders**

1. We use a consistent batch size of 32.
2. For all models, we try to consistently use six layers, each with 512 hidden channels. For tensor-based equivariant GNNs (TFN, MACE), we reduced the number of layers and hidden channels to fit 80GB of GPU memory on one NVIDIA A100 GPU.
3. For each encoder and featurisation baseline, we search over learning rates: $0.00001, 0.0001, 0.0003, 0.001$ and select the best based on the validation performance on the fold classification task.
4. For each finetuning configuration, we conduct a small sweep over learning rates: $0.0001, 0.0003, 0.00001$ and perform model selection based on the validation performance.
5. We train all models to convergence or to a maximum of 24 hours on a single A100 GPU.

**Output Heads**

1. All primary output heads use a three-layer MLP with 512 as the hidden dimension.
2. For all auxiliary tasks we use a two-layer MLP with 128 as the hidden dimension.
3. For all structure denoising tasks we use a two-layer SE(3) equivariant noise predictor network (Section D.9). The message and distance MLPs each consist of two layers of 128 hidden units.
4. For each encoder and featurisation, we search over decoder dropout: $0.0, 0.1, 0.3, 0.5$, and select the best based on the validation performance on the Fold Classification task.

## E DOCUMENTATION FOR DATASETS

Below, we provide detailed documentation for each dataset included in our benchmark, summarised in Table 6. Each dataset will be made available for download in processed and raw forms from Zenodo upon publication. Note that, for all datasets, we authors bear all responsibility in case of any violation of rights regarding the usage of such datasets, whether they were compiled from existing sources or curated from scratch.

### E.1 CATH - INVERSE FOLDING

This is a common protein engineering task where the goal is to recover an amino acid sequence given a structure up to backbone completeness. Formally, this is a node-level classification task where the

model learns a mapping for each residue $r_i$ to an amino acid type $y \in \{1, \dots, n\}$, where $n$ is the vocabulary size ($n = 20$ for the canonical set of amino acids).

Note that inverse folding is a generic task that can be applied to any dataset. In the literature, it is commonly evaluated on the CATH dataset compiled by Ingraham et al. (2019). We additionally use inverse folding on AlphaFoldDB as a pretraining task.

- **Motivation** Several generative methods for protein design produce backbone structures that require the design of an associated sequence. As a result, inverse folding is an important part of *de novo* design pipelines for proteins.
- **Collection** For this dataset, we adopt the commonly-used CATH dataset originally compiled by Ingraham et al. (2019).
- **Composition** The dataset consists of protein structures randomly split into training, validation and test sets such that proteins in different sets do not share the same CATH topology classification (i.e., CAT code).
- **Hosting** A preprocessed version of the dataset can be downloaded from the benchmark's Zenodo data record.
- **Licensing** We have released a preprocessed version of the dataset under a Creative Commons Attribution 4.0 International license. The original dataset is available under a Creative Commons Attribution 4.0 International license at `http://cathdb.info`.
- **Maintenance** We will announce any errata discovered in or changes made to the dataset using the benchmark's GitHub repository.
- **Uses** This dataset can be used for multilabel node classification tasks where a model learns a mapping for each residue $r_i$ to an amino acid type $y \in \{1, \dots, n\}$, where $n$ is the vocabulary size (e.g., $n = 20$ for the canonical set of amino acids).
- **Metric** Perplexity.

### E.2 MASIF-SITE - PPI SITE PREDICTION

This task is a node-level binary classification task where the goal is to predict whether or not a residue is involved in a protein-protein interaction interface.

- **Motivation** Identifying protein-protein interaction (PPI) sites has important applications in developing improved protein-protein interaction networks and docking tools, providing biological context to guide protein engineering and target identification in drug discovery campaigns (Jamasb et al., 2021).
- **Collection** We adopt the dataset of experimental structures curated from the PDB by Gainza et al. (2020) and retain the original splits, though we modify the labelling scheme to be based on inter-atomic proximity (3.5 Å), which can be user-defined, rather than solvent exclusion.
- **Composition** The dataset is curated from the PDB by preprocessing such as the presence of one of the seven specified ligands (e.g., ADP or FAD), clustering based on 30% sequence identity and random subsampling. It contains 1,459 structures, which are randomly assigned to training (72%), validation (8%) and test set (20%). 12 (Å) radius patches were extracted from the generated structures and a patch labelled as part of a binding pocket if its centre point was < 3 (Å) away from an atom of the corresponding ligand.
- **Hosting** The original dataset is made available by the authors on Zenodo.
- **Licensing** We have released a preprocessed version of the dataset under a Creative Commons Attribution 4.0 International license. The original dataset is available under an Apache 2.0 license at `https://github.com/LPDI-EPFL/masif/blob/master/LICENSE`.
- **Maintenance** We will announce any errata discovered in or changes made to the dataset using the benchmark's GitHub repository.
- **Uses** This dataset can be used for binary node classification tasks where the goal is to predict whether or not a residue is involved in a protein-protein interaction interface.
- **Metric** AUPRC.

### E.3    CCPDB - METAL BINDING SITE PREDICTION

This is a binary node classification task where each residue is mapped to a label $y \in \{0, 1\}$ indicating whether the residue (or its constituent atoms) is within 3.5 (Å) of a user-defined metal ion or ligand heteroatom, respectively.

- **Motivation** Several proteins coordinate transition metal ions to carry out their functions. As such, predicting the binding sites of metal ions can elucidate the role of metal binding on protein function.

- **Collection** The dataset is constructed from experimental structures curated from the PDB, where binding site assignments for each residue are computed on-the-fly. While the benchmark supports this task on arbitrary subsets of the PDB and ligands, we provide the Zinc-binding dataset from Dürr et al. (2023) specifically for this task.

- **Composition** The dataset is constructed by sequence-based clustering of the PDB at 30% sequence identity to remove sequence and structural redundancy. Clusters with a member shorter than 3000 residues, containing at least one zinc atom with resolution better than 2.5 (Å) determined by x-ray crystallography and not containing nucleic acids are used to compose the dataset. If multiple structures fulfill these criteria, the highest resolution structure is used. The train (2,085) / validation (26) / test (59) splits are constructed such that proteins in the validation and test sets have no partial overlap with any protein in the training data.

- **Hosting** A preprocessed version of the dataset can be downloaded from the benchmark's Zenodo data record.

- **Licensing** We have released a preprocessed version of the dataset under a Creative Commons Attribution 4.0 International license. The original dataset is freely available without a license at https://academic.oup.com/database/article/doi/10.1093/database/bay142/5298333#130010908.

- **Maintenance** We will announce any errata discovered in or changes made to the dataset using the benchmark's GitHub repository.

- **Uses** This dataset can be used for binary node classification tasks where each residue is mapped to a label $y \in \{0, 1\}$ indicating whether the residue (or its constituent atoms) is within 3.5 (Å) of a user-defined metal ion or ligand heteroatom, respectively.

- **Metric** Accuracy.

### E.4    PTM - POST-TRANSLATIONAL MODIFICATION SITE PREDICTION

We frame prediction of post-translational modification (PTM) sites as a multilabel classification task where each residue is mapped to a label $y \in \{1, \ldots, 13\}$ distinguishing between modifications on different amino acids (e.g. phosphorylation on S/T/Y and N-linked glycosylation on N).

- **Motivation** Identifying the exact sites where post-translational modifications (PTMs) occur is essential for understanding protein behaviour and designing targeted therapeutic interventions.

- **Collection** We adopt a dataset of 48,811 AlphaFold2-predicted structures curated by Yan et al. (2023), where each structure contains the PTM metadata necessary to construct residue-wise site prediction labels.

- **Composition** The dataset is split into training (43,907), validation (2,393) and test (2,511) sets based on 50% sequence identity and 80% coverage. In total, there are 240,090 PTMs present in the dataset compared to 3,391,208 residues where PTMs could be possible but are not present. The most common PTMs are phosphorylations on serine (93,734) and N-linked glycosylation at asparagine (59,143) which together account for around 70% of the PTMs.

- **Hosting** A preprocessed version of the dataset can be downloaded from the benchmark's Zenodo data record.

- **Licensing** We have released a preprocessed version of the dataset under a Creative Commons Attribution 4.0 International license. The original dataset is available under a Creative Commons Attribution 4.0 International license at https://zenodo.org/record/7655709.

- **Maintenance** We will announce any errata discovered in or changes made to the dataset using the benchmark's GitHub repository.
- **Uses** This dataset can be used for multilabel node classification tasks where each residue is mapped to a label $y \in \{1, \ldots, 13\}$ distinguishing between modifications on different amino acids (e.g., phosphorylation on S/T/Y and N-linked glycosylation on N).
- **Metric** ROC-AUC.

### E.5 FOLD - FOLD PREDICTION

This is a multiclass graph classification task where each protein, $\mathcal{G}$, is mapped to a label $y \in \{1, \ldots, 1195\}$ denoting the fold class.

- **Motivation** The utility of fold prediction is that it serves as a litmus test for the ability of a model to distinguish different structural folds. It stands to reason that models that perform poorly on distinguishing fold classes likely learn limited or low-quality structural representations.
- **Collection** We adopt the fold classification dataset originally curated from SCOP 1.75 by (Hou et al., 2017).
- **Composition** This dataset provides three different test sets stratified based on topological similarity: Fold, in which proteins originating from the same superfamily are absent during training; Superfamily, in which proteins originating from the same family are absent during training; and Family, in which proteins from the same family are present during training.
- **Hosting** A preprocessed version of the dataset can be downloaded from the benchmark's Zenodo data record.
- **Licensing** We have released a preprocessed version of the dataset under a Creative Commons Attribution 4.0 International license. The original dataset is available under a Creative Commons Attribution 4.0 International license at `https://academic.oup.com/bioinformatics/article/34/8/1295/4708302`.
- **Maintenance** We will announce any errata discovered in or changes made to the dataset using the benchmark's GitHub repository.
- **Uses** This dataset can be used for multilabel graph classification tasks where each protein, $\mathcal{G}$, is mapped to a label $y \in \{1, \ldots, 1195\}$ denoting the fold class.
- **Metric** Micro-averaged accuracy.

### E.6 GO - GENE ONTOLOGY PREDICTION

This is a multilabel classification task, assigning functional Gene Ontology (GO) annotation to structures. GO annotations are assigned within three ontologies: biological process (BP), cellular component (CC) and molecular function (MF).

- **Motivation** Predicting protein function in the form of functional annotations such as gene ontology (GO) terms has important applications in protein analysis and engineering, providing researchers with the ability to cluster functionally-related structures or to guide protein generation methods to design new proteins with desired functional properties.
- **Collection** We adopt the dataset of experimental structures originally curated from the PDB by Gligorijević et al. (2021).
- **Composition** We retain the original multi-cutoff based dataset splits proposed by (Gligorijević et al., 2021), with cutoff at 30% sequence similarity.
- **Hosting** A preprocessed version of the dataset can be downloaded from the benchmark's Zenodo data record.
- **Licensing** We have released a preprocessed version of the dataset under a Creative Commons Attribution 4.0 International license. The original dataset is available under a Creative Commons Attribution 4.0 International license at `https://www.nature.com/articles/s41467-021-23303-9`.

- **Maintenance** We will announce any errata discovered in or changes made to the dataset using the benchmark's GitHub repository.
- **Uses** This dataset can be used for multilabel graph classification tasks, assigning a functional Gene Ontology (GO) annotation to protein structures.
- **Metric** $F_{max}$ score.

### E.7 EC REACTION - REACTION CLASS PREDICTION

This is a multiclass graph classification task where each protein, $\mathcal{G}$, is mapped to a label $y \in \{1, ..., 384\}$ denoting which class of reactions a given protein catalyzes; all four levels of the EC assignment are employed to define the reaction class label.

- **Motivation** As proteins' reaction classifications are based on their enzyme-catalyzed reaction according to all four levels of the standard Enzyme Commission (EC) number, methods that predict such classifications can help elucidate the function of newly-designed proteins as they are developed.
- **Collection** We adopt the reaction class prediction dataset originally curated from the PDB by Hermosilla et al. (2020).
- **Composition** The dataset is split on the basis of sequence similarity using a 50% threshold.
- **Hosting** A preprocessed version of the dataset can be downloaded from the benchmark's Zenodo data record.
- **Licensing** We have released a preprocessed version of the dataset under a Creative Commons Attribution 4.0 International license. The original dataset is available under an MIT license at `https://github.com/phermosilla/IEConv_proteins/blob/master/LIC ENSE`.
- **Maintenance** We will announce any errata discovered in or changes made to the dataset using the benchmark's GitHub repository.
- **Uses** This dataset can be used for multilabel graph classification tasks where each protein, $\mathcal{G}$, is mapped to a label $y \in \{1, ..., 384\}$ denoting which class of reactions a given protein catalyzes.
- **Metric** Accuracy.

### E.8 TDC - ANTIBODY DEVELOPABILITY PREDICTION

Therapeutic antibodies must be optimised for favourable physicochemical properties in addition to target binding affinity and specificity to be viable development candidates. Consequently, we frame prediction of antibody developability as a binary graph classification task indicating whether a given antibody is developable.

- **Motivation** From a benchmarking perspective, predicting the developability of a given antibody is important as it enables targeted performance assessment of models on a specific (immunoglobulin) fold, providing insight into whether general-purpose structure-based encoders can be applicable to fold-specific tasks.
- **Collection** We adopt the antibody developability dataset originally curated from SabDab (Dunbar et al., 2014) by Chen et al. (2020).
- **Composition** This dataset contains 2,426 antibodies that have both sequences and PDB structures available, where each example contains both a heavy chain and a light chain with resolution < 3 (Å). Labels are based on thresholding the developability index (DI) ((Lauer et al., 2012)) as computed by BIOVIA's platform ((Systèmes, 2016)), which relies on an antibody's hydrophobic and electrostatic interactions.
- **Hosting** A preprocessed version of the dataset can be downloaded from the benchmark's Zenodo data record.
- **Licensing** We have released a preprocessed version of the dataset under a Creative Commons Attribution 4.0 International license. The original dataset is available under a Creative Commons Attribution 3.0 Unported license at `https://tdcommons.ai/single_pred_tasks /develop/#sabdab-chen-et-al`.

- **Maintenance** We will announce any errata discovered in or changes made to the dataset using the benchmark's GitHub repository.
- **Uses** This dataset can be used for binary graph classification tasks indicating whether a given antibody is developable.
- **Metric** AUPRC.

