# OpenReview forum: "Evaluating Representation Learning on the Protein Structure Universe"
_ICLR.cc/2024/Conference — ICLR 2024 poster_

### Official Review · Reviewer_keJ2 · 2023-10-20

**Soundness:** 3 good
**Presentation:** 4 excellent
**Contribution:** 3 good
**Rating:** 8
**Confidence:** 4

**Summary:**

The authors introduce a large suite of benchmarks for evaluating learned embeddings of proteins. Included are node-level evaluations (at the level of individual residues, e.g. inverse folding, metal binding site prediction) and graph-level evaluations (at the level of the entire protein, e.g. fold classification). The authors also provide a number of software tools, including dataloaders for various pretraining tasks. Finally, they evaluate selected architectures on various pretraining task/benchmark combinations.

**Strengths:**

New benchmarks are always welcome, and there is also great value in consolidating existing benchmarks. This paper does that well, as the selection here is quite broad, with good coverage. I appreciate the inclusion of both node-level and graph-level evaluations. Within each category, each choice of benchmark is accompanied by a specific rationale, which is also a strength. The documentation of the codebase also seems clear and easy to follow.

**Weaknesses:**

It would have been nice to see baselines for all of the downstream tasks in the benchmark using the tools in this software suite (or, lacking that, even scores copied from their respective papers if needs be). Certain tasks like "reaction class prediction" are currently missing. Part of the value-add here is the ease of running experiments on all of these tasks, and that isn't currently demonstrated in the current version of the manuscript. Also, consolidated baseline scores are useful sanity checks for reproduction experiments down the line.

Miscellaneous stuff:

- Please provide details about the confidence intervals in Table 2.
> whereas pLDDT prediction and structure denoising benefit invariant models the most
- I don't really understand what the basis of this claim (^) is. All four models do approximately as well in the pLDDT column, e.g.

**Questions:**

In Table 3, inverse folding (a downstream node-level task) is listed alongside four other tasks explicitly identified as pretraining tasks above. Is this intentional?

Are the experiments in Table 2 at all realistic? I don't think these tasks ever be attempted using models with no pretraining at all in practice. Also, the ESM model without any pretraining can hardly be called an ESM model at all.

Did you consider adding any restrictions on the interfaces between model embeddings and the downstream tasks? If the goal is to evaluate the embeddings themselves, it seems to me like there should be an attempt to standardize e.g. the size of the task-specific MLPs used.

---

> ### Author Response · Authors · 2023-11-18
> **Response to Reviewer keJ2**
>
> We thank the reviewer for their careful consideration of our work. We are especially glad the reviewer found the coverage of our benchmark to be broad and the codebase to be well documented -- we have put a lot of effort into the engineering!
>
> ---
> > It would have been nice to see baselines for all of the downstream tasks in the benchmark using the tools in this software suite...
>
> Please stay tuned for the updated manuscript containing more downstream results. We will upload this ASAP.
>
> ---
> > Please provide details about the confidence intervals in Table 2.
>
> We report the standard deviation over three runs across different seeds. We will mention this in the revised paper.
>
> ---
> > In Table 3, inverse folding (a downstream node-level task) is listed alongside four other tasks explicitly identified as pretraining tasks above. Is this intentional?
>
> Thank you for raising this ambiguity. We use the inverse folding task both as a downstream evaluation task (on the CATH-derived dataset by [Ingraham et al](https://www.mit.edu/~vgarg/GenerativeModelsForProteinDesign.pdf)), which is commonly used in the field for this task. We also investigate an inverse folding pre-training objective on the afdb_rep_v4 dataset which we subsequently finetune on the CATH dataset.
>
> ---
> > Are the experiments in Table 2 at all realistic? I don't think these tasks ever be attempted using models with no pretraining at all in practice.
>
> We thank the reviewer for raising this point. We generally agree with the reviewer on this point. These tasks are generally (but not exclusively) attempted with pre-trained models. However, they serve as an important point of reference for understanding the performance gains through pre-training, and investigating whether simpler strategies can achieve comparable results.
>
> ---
> > Also, the ESM model without any pretraining can hardly be called an ESM model at all.
>
> We in fact used a frozen pre-trained ESM2 650M parameter model. We will amend the caption for Table 2.
>
> We chose the 650M variant of ESM2 because this is the parameter count/scale at which the ESM2 authors observed significant improvements in the model’s ability at structure-related tasks. Our results further reinforce their observation – we find that combining ESM2 650M with our structural featurisation yields extremely strong results on Fold classification tasks.
>
> ---
> > Did you consider adding any restrictions on the interfaces between model embeddings and the downstream tasks? If the goal is to evaluate the embeddings themselves, it seems to me like there should be an attempt to standardize e.g. the size of the task-specific MLPs used.
>
> We used task-specific MLPs of a consistent size (3 layers with the same width as the GNN encoder, with skip connections), tuning the dropout on the FoldClassification task. We fixed a consistently large hidden dimensionality across GNN encoders (at most 512). See Appendix B5 for more details.

---

> > ### Comment · Reviewer_keJ2 · 2023-11-21
> >
> > Thanks for the reply. I look forward to the additional evaluations.

---

### Official Review · Reviewer_foh9 · 2023-10-31

**Soundness:** 3 good
**Presentation:** 4 excellent
**Contribution:** 3 good
**Rating:** 6
**Confidence:** 5

**Summary:**

The authors have introduced a novel framework aimed at curating datasets sourced from public repositories like PDB. Their objective is to construct benchmark datasets that facilitate the evaluation of protein structure representation. This framework takes into account various settings, including different backbone models such as various GNN-based models and diverse feature representations, to enhance our understanding of protein structure presentation. Additionally, the authors explore a wide range of pretraining tasks, including sequence/structure denoising and inverse folding, before subjecting these models to a battery of diverse downstream tasks, operating at the Alpha carbon, residue, or overall protein levels.

To support this work, the authors have generously shared an anonymous GitHub link containing scripts for data preprocessing, which enables the creation of datasets for both pretraining and benchmarking. This contribution is particularly noteworthy as it addresses a critical gap in the field of protein structure representation learning. Historically, the lack of systematically curated benchmarking datasets covering aspects such as featurization, pretraining, and downstream task evaluation has hindered progress. The absence of such a framework has made it exceedingly challenging to compare or reproduce research findings in this specialized domain.

However, there are some concerns regarding the paper's completeness. Notably, the authors have outlined a set of downstream tasks in Figure 1, yet a significant portion of these tasks remains unreported in the results section.

**Strengths:**

The unveiling of this framework, designed for assembling public datasets in order to generate pretraining and downstream benchmark datasets for the study of protein structure representation, is a noteworthy development.

The examination of how pretraining and featurization affect various downstream architectures and tasks proves to be engaging and insightful.

The timeliness and significance of this research topic cannot be understated, as it addresses the pressing need for a standardized framework that enables the comparison of various state-of-the-art methods using the same benchmark datasets.

**Weaknesses:**

Addressing the Issue of Potential Leakage:

Efforts to mitigate potential data leakage are crucial to ensuring the integrity of benchmarking results, as such leakage could introduce misleading elements into the research findings. Have you considered the removal of overlapping sequences between the pretraining datasets and the downstream testing datasets to further safeguard against such issues?

Expanding Featurization Methods:

In terms of featurization, the paper seems to primarily focus on simple feature extraction methods. It might be valuable if the authors explored the integration of the following additional featurization techniques:

    Incorporating 3D presentations, such as Uni-Mol (available at https://github.com/dptech-corp/Uni-Mol).
    Utilizing pretrained models trained on 2D data, like ESM-1b and ESM-2. Considering that Table 1 showcases results using ESM as a backbone model, which produced superior outcomes on the fold dataset, it could be beneficial to include the results for other ESM models in the table.

Diverse Downstream Tasks:

Table 1 predominantly presents results for two specific tasks. However, it appears that several other downstream tasks have not been included in this experiment. It is worth noting that many of these unreported downstream tasks are of substantial importance and are conspicuously absent from the paper. Is there a particular reason for not considering these tasks, and could they potentially be included to provide a more comprehensive view of the research findings?

**Questions:**

1. Could you please address the potential leakage as discussed in the weakness comments?

2. Could you please consider additional featurization as pointed out in the weakness comments?

3. Could you please complete the experiments and provide the results with the downstream tasks that are missing in the paper?

---

> ### Author Response · Authors · 2023-11-18
> **Response to Reviewer foh9 (Part 1/n)**
>
> Thank you for your constructive feedback and for finding our work timely and significant!
>
> > Efforts to mitigate potential data leakage are crucial to ensuring the integrity of benchmarking results, as such leakage could introduce misleading elements into the research findings. Have you considered the removal of overlapping sequences between the pretraining datasets and the downstream testing datasets to further safeguard against such issues?
>
> Thank you for raising this point. Please consider our counter argument against why using a part of the downstream test dataset for pre-training neural networks is okay.
>
> We believe that utilizing a portion of the downstream data, **without labels**, to find a suitable model initialization that improves performance on the end task is acceptable in our setting. Our primary goal with using a large corpus of AlphaFold DB structures is to boost downstream task accuracy, and not directly use the label information, which contains key details. This is in line with models in natural language processing like BERT (and even SciBERT), the full Wikipedia datasets were utilized for BERT pre-training without carefully removing any overlap with fine-tuning sets.
>
> Now, protein data may be a special case since exactly identical sequences and structures exist across datasets, unlike in language data. We somewhat mitigate this by using the non-redundant set of AFDB which are predicted structures which consider idealised geometry. There is still a semantic gap between crystal structures from the PDB (our downstream tasks) and AFDB. They are never identical.
>
> However, we do agree that when it comes to the domain generalization properties of pre-training models, it might be better to explicitly exclude certain samples in the pre-training phases.
>
> Another counterargument is that it’s fairly straightforward to obtain and utilize predicted protein structure information from tools like AlphaFold during pre-training in real-world settings. Researchers working on real-world protein modelling and design problems are actively starting to use AlphaFold-predicted structures to augment their datasets and are obtaining powerful gains from it. Two notable examples include:
> - [Learning inverse folding from millions of predicted structures](https://www.biorxiv.org/content/10.1101/2022.04.10.487779v1), Hsu et al. (ICML oral, 2022)
> - [RosettaFold2](https://www.biorxiv.org/content/10.1101/2023.05.24.542179v1), which uses AlphaFold-predicted structures to train the latest RosettaFold version.
>
> In sum, as long as the pre-training data does not include labels and is consistent across models, we believe it is a reasonably fair methodology for improving downstream performance, and the particular scenario of AlphaFold-predicted structures is already being used in real-world scenarios to the best of our knowledge.
>
> ---
> > It might be valuable if the authors explored the integration of the following additional featurization techniques: Incorporating 3D presentations, such as Uni-Mol
>
> Uni-Mol is focused on protein binding pocket and ligand (small molecule) interactions, which is **not directly applicable** to our setting. Quoting the repository, “The framework comprises two models: a molecular pretraining model that has been trained using 209M molecular 3D conformations, and a pocket pretraining model that has been trained using 3M candidate protein pocket data.”
>
> ProteinWorkshop focuses on protein-only representation learning and self-supervised pre-training at the node and graph level. Critically, we are interested in tasks that consider the **entire** protein structure, as opposed to the setting of considering a cropped patch (the protein pocket) and its interaction with a ligand.
>
> In the future, it would be interesting to extend our framework to biomolecular complexes, as the PDB remains the main data source. (See [related response](https://openreview.net/forum?id=sTYuRVrdK3&noteId=Ha3Lo64kpL) to Reviewer W2RU.)

---

> ### Author Response · Authors · 2023-11-18
> **Response to Reviewer foh9 (Part 2/n=2)**
>
> > Utilizing pretrained models trained on 2D data, like ESM-1b and ESM-2....it could be beneficial to include the results for other ESM models in the table.
>
> At present, we chose the 650M variant of ESM2 because this is the parameter count/scale at which the ESM2 authors observed significant improvements in the model’s ability at structure-related tasks. (See Figure 1 in [the Science paper](https://www.science.org/doi/10.1126/science.ade2574) introducing ESM2.) Our results further reinforce their observation – we find that combining ESM2 650M with our structural featurisation yields extremely strong results on Fold classification tasks.
>
> Our goal with the benchmark was to demonstrate the utility of our ProteinWorkshop framework and to provide new insights from fairly comparing standard architecture choices. It should be fairly straightforward to swap the exact version of the ESM language model used for featurisation within ProteinWorkshop, but we believe we have already used one of the best ESM variant for structural tasks. While we cannot be certain without having done the experiment, it is a reasonable guess that a variant of ESM2 with fewer parameters, or the older generation of ESM1b models, will not perform any better than ESM2 650M nor would this change the outcome of our benchmarks.
>
> We included ESM2 as a reference, but our focus is on structure-based geometric GNN models and scaling them up.
>
> ---
>
> > Table 1 predominantly presents results for two specific tasks. However, it appears that several other downstream tasks have not been included in this experiment. It is worth noting that many of these unreported downstream tasks are of substantial importance and are conspicuously absent from the paper. Is there a particular reason for not considering these tasks, and could they potentially be included to provide a more comprehensive view of the research findings?
>
> Thank you for raising this point. Please stay tuned for the updated manuscript containing more downstream results. We will upload this ASAP.

---

> > ### Comment · Reviewer_foh9 · 2023-11-22
> > **Still missing downstream tasks in Figure 1**
> >
> > Thank you for providing your responses to my comments. I have carefully reviewed both the comments from other reviewers and the author's response. However, I maintain the perspective that the paper remains incomplete. Specifically, the handling of leakage lacks thorough attention, essential baselines related to the ESM models have been ignored, and a critical shortcoming is the absence of a significant portion of the promising downstream tasks highlighted in Figure 1 within the experimental framework.
> >
> > Given that this paper focuses on a benchmark framework, completeness is crucial for its utility within the research community. It is my belief that addressing these concerns would significantly enhance the overall quality and applicability of the paper.
> >
> > Best regards,

---

> > > ### Author Response · Authors · 2023-11-22
> > >
> > > Thanks for engaging with us and the further comments, we are extremely grateful to the reviewer for helping us to improve our work.
> > >
> > > > Specifically, the handling of leakage lacks thorough attention
> > >
> > > We must respectfully disagree with the reviewer on this point. There is no possibility of label leakage in our pre-training tasks.
> > >
> > > It is conventional practice to pre-train on large datasets (*without* the use of label information) and finetune on downstream tasks. In fact, the ESM family of pre-trained protein language models has set this precedent already in the protein modelling community. We do not see our pretraining on AFDB as any different from what has been done in prominent works such as ESM, ProteinBERT, GearNet, and others. Petraining machine learning models on large scale datasets has become standard practice in real-world protein engineering and design campaigns and we would like to make our work relevant for practitioners. Given the reviewer believes additional ESM results would be valuable (addressed below), we make the case that our experimental setup should be considered no differently. We are unable to intervene on the sequences on which ESM was trained which will contain many more related sequences to the test examples (and almost certainly more exact sequence matches).
> > >
> > > Finally, to address the practicality of this. Removing similar structures as the reviewer suggests would necessitate repeating each of the pre-training configurations on a slightly modified dataset for each downstream task. This would require unreasonable amounts of resources which we do not believe are justified given the objections we have stated and consensus  in the field.
> > >
> > > > Essential baselines related to the ESM models have been ignored
> > >
> > > We would like to reiterate that our focus is on structure-based models, and the effectiveness of structure-based pre-training/auxiliary tasks. We include ESM as a reference point but the focus is on structure-based models. It is not clear what benefit examining other ESM based models will add to this objective.
> > >
> > > > a critical shortcoming is the absence of a significant portion of the promising downstream tasks highlighted in Figure 1 within the experimental framework.
> > >
> > > Please bear with us; we are working on adding as many evaluations as possible to the manuscript before the close of the rebuttal period. Please note the significant resources required to obtain these results across all the models, featurisations, and auxilliary tasks we consider. If accepted, we are committed to obtaining a full complement of baselines by the camera-ready deadline.

---

> > > > ### Comment · Area_Chair_K82L · 2023-12-02
> > > > **Does the response address your concerns?**
> > > >
> > > > @Reviewer foh9,
> > > >
> > > > I would appreciate it if you could review the response and adjust your feedback (and rating) as necessary.
> > > >
> > > > AC

---

### Official Review · Reviewer_5uRv · 2023-11-02

**Soundness:** 3 good
**Presentation:** 3 good
**Contribution:** 2 fair
**Rating:** 5
**Confidence:** 5

**Summary:**

This paper focuses on the task of protein structure representation learning and aims to provide a robust and standardized benchmark for this task.

In this paper, the authors provide different pretraining datasets, downstream datasets, pretraining tasks, auxiliary tasks, featurisation schemes, and model architectures. They cover most of the widely used training strategies, datasets, and model architectures.

In addition, the authors run experiments using the provided code base and provide some observations and insights.

**Strengths:**

This paper is well-written and easy to follow.

The provided datasets, GNN models, and training strategies are comprehensive.

**Weaknesses:**

1. In addition to datasets etc, I think a good benchmark should also provide experimental results with well-searched hyperparameters. In such case, future researchers can directly take results for a fair comparison.
 - However, in the current version, the authors didn’t provide results on all downstream tasks.
 - In addition, I am not sure whether the hyperparameters are well-searched, since the best results reported here are still worse than some existing methods. For example, the best results on Fold (considering both with and without auxiliary tasks) are still worse than ProNet [1] and CDConv [2] which don’t use any auxiliary tasks.

2. Do ESM results in the table use pre-trained ESM weights?

3. Some other pretraining strategies are used in Geom3D [3] and ESM-GearNet [4].

[1] Learning Hierarchical Protein Representations via Complete 3D Graph Networks.
[2] Continuous-Discrete Convolution for Geometry-Sequence Modeling in Proteins.
[3] Symmetry-Informed Geometric Representation for Molecules, Proteins, and Crystalline Materials.
[4] A Systematic Study of Joint Representation Learning on Protein Sequences and Structures

**Questions:**

See weaknesses

---

> ### Author Response · Authors · 2023-11-18
> **Response to Reviewer 5uRv (Part 1/n)**
>
> We thank the reviewer for their feedback and constructive comments. We are glad the reviewer found our benchmark comprehensive and the manuscript clear.
>
> > However, in the current version, the authors didn’t provide results on all downstream tasks.
>
> Please stay tuned for the updated manuscript containing more downstream results. We will upload this ASAP.
>
> ---
> > In addition, I am not sure whether the hyperparameters are well-searched, since the best results reported here are still worse than some existing methods. For example, the best results on Fold (considering both with and without auxiliary tasks) are still worse than ProNet and CDConv
>
> Given the large number of models and featurisation schemes, we did try our best to do a consistent and fair hyperparameter search. See Appendix B5 for details -- essentially, we used the fold classification task to select the best learning rate and dropout per model and featurisation scheme. We fixed a consistently large hidden dimensionality across models, as we also wanted to focus on scaling model size and dataset size via pre-training.
>
> While our best performing models do not outperform the best reported results on Fold, we have obtained these results in a consistent and fair experimental setup. Our goal was to demonstrate the utility of our benchmarking framework and uncover the impact of architectural considerations such as featurisation schemes, geometric GNN models, and pre-training/auxiliary tasks. We are confident that the insights from our benchmarking study will subsequently help us build models that outperform the state of the art across both node and graph level representation learning tasks.
>
> Finally, it is worth stating that both CDConv and ProNet are multi-scale architectures with a **highly specialised design tailored to graph/global-level prediction tasks**. It is not trivial to generalise such architectures for node-level tasks, unlike the suite of generic as well as protein-specific geometric GNNs considered in this work.
>
> In summary, while the subject of this particular paper is to introduce ProteinWorkshop as a platform for evaluating protein representation learning, we are now actively working on building state of the art models for node and graph level protein tasks based on the insights gained!
>
> ---
> > Do ESM results in the table use pre-trained ESM weights?
>
> Yes, we use a frozen pre-trained ESM2 650M parameter model. We apologize for the oversight and have added this point to the paper. We chose the 650M variant of ESM2 because this is the parameter count/scale at which the ESM2 authors observed significant improvements in the model’s ability at structure-related tasks. Our results further reinforce their observation – we find that combining ESM2 650M with our structural featurisation yields extremely strong results on Fold classification tasks.

---

> ### Author Response · Authors · 2023-11-18
> **Response to Reviewer 5uRv (Part 2/n=2)**
>
> > Some other pretraining strategies are used in Geom3D [3] and ESM-GearNet [4].
>
> Re. Geom3D and ESM-GearNet: We will cite these papers in the revised version. Do note that Geom3D strategies seem to be specialised for small molecules (as stated in section 4.7 of their paper) but some of them are reasonably close to those studied by us, distance/angle prediction.
>
> Let us provide further justification of how we selected our pre-training tasks.
>
> We focussed on pre-training tasks that roughly fall under the category of denoising (eg. sequence, coordinates). We were particularly interested in self-supervised objectives that were:
> 1. Extremely scalable, so as to pre-train on the large scale AlphaFold Protein Structure Database (AFDB) of 2.4M structures; and
> 2. Train protein representations at the fine-grained node level, so as to be general-purpose across the downstream tasks considered.
>
> Naturally, we would have liked to continue exploring more pre-training strategies, but could not do so due to the significant cost of running each pre-training experiment on 2.4M AFDB structures (total cost of GPU hours roughly being $50,000). Notably, the structural clustering of AFDB we use results in a dataset approx. 3x larger than what is used for pre-training GearNet and its variants. Thus, the scalability of the pre-training objectives was a critical consideration in selecting them for our benchmarking study.
>
> We did not yet consider other possible tasks that have also been used in the literature, such as contrastive learning and generative modelling-inspired objectives. Most of these objectives are (1) computationally heavier and more cumbersome to set up than corruption-type objectives, making them harder to scale up, and (2) only train protein representations for the global/graph level and do not operate at the node level.
>
> We would also like to highlight that, ultimately, ProteinWorkshop is a platform through which we certainly hope to integrate future pre-training objectives such as those from complementary studies like Geom3D. The goal of the benchmarking study in this paper was to **demonstrate the utility of our framework** and making pre-training on the largest non-redundant corpus of protein structures accessible to the community. We believe this makes ProteinWorkshop an ideal framework to advance self-supervised protein representation learning in the future.

---

> > ### Comment · Reviewer_5uRv · 2023-11-22
> >
> > Thanks for the authors' response.
> >
> > Overall, I agree with the contribution.
> > However, my main concern **results for all downstream tasks are not currently available** is not well addressed. As someone in this research area, I really hope this tool can be used to fairly benchmark different methods and I can directly compare to the results listed in this paper in my future research.
> >
> > Looking forward to further experimental results, for now, I will keep my score as 5.

---

> > > ### Comment · Area_Chair_K82L · 2023-12-02
> > > **Does the latest version address your concern about results over downstream tasks?**
> > >
> > > Reviewer 5uRv,
> > >
> > > I would appreciate it if you could review the lastest version and adjust your feedback (and rating) as necessary.
> > >
> > > AC

---

### Official Review · Reviewer_W2RU · 2023-11-07

**Soundness:** 3 good
**Presentation:** 3 good
**Contribution:** 2 fair
**Rating:** 6
**Confidence:** 3

**Summary:**

This work presents an open benchmark for evaluating protein structure representation-learning methods. The benchmark includes a diverse set of pre-training methods, downstream tasks, and corpora and includes experimental and predicted protein structures. The structure-based pre-training and fine-tuning datasets and tasks emphasize tasks that enable structural annotation.

**Strengths:**

1. Modular benchmark enabling rapid evaluation of protein representation learning methods across various tasks, models, representations, and pre-training setups.

2. Analysis of model performance across these different representations and architectures.

3. Using auxiliary tasks to improve the performance of both invariant and equivariant models.

4. Providing tools and procedures for training and evaluating models.

**Weaknesses:**

1. The work is missing an explanation of the limitations of the featurization schemes and pre-training tasks.

2. Would be beneficial to include a discussion about the generalizability of the benchmark results
to the overall protein structure space, and how this translates to proteins not included in the current dataset.

3. Missing a discussion about how geometric models may be improved to surpass sequence-based models.

4. Missing information about the ease of use of the tools, and details about the computational resources required for using the benchmark.

5. Missing (i) aggregation of methods for improving model performance; and (ii) computation of uncertainties in evaluations.

**Questions:**

Can the work be adapted for other biological macromolecules beyond protein structures?
See, for example, the recent reference:
Performance and structural coverage of the latest, in-development AlphaFold model, 2023.
Predicting structure of proteins, nucleic acids, small molecules, ions, and modified residues. Providing a quantitative benchmark, and improving accuracy of protein-ligand structure prediction, protein-DNA and protein-RNA interface structure prediction, and protein-protein interfaces.

---

> ### Author Response · Authors · 2023-11-18
> **Response to Reviewer Reviewer W2RU (Part 1/n)**
>
> Thank you for your constructive comments on our benchmarking framework as well as analysis! We will improve the work based on all your suggestions which were very useful.
>
> > The work is missing an explanation of the limitations of the featurization schemes and pre-training tasks.
>
> Thank you for the suggestion – we will now include a paragraph each on the limitations of current featurization schemes as well as pre-training tasks in the revised version (which we are preparing with results on new datasets).
>
> **Limitations of featurization schemes:**
> - All of our featurization schemes are at the residue level, with the position of the alpha Carbon atom being assigned the position of a particular node. This is conventional practice in the field due to efficiency, as all atom representations require very high GPU memory/make models extremely slow.
> - The extent to which each scheme is a ‘complete’ representation of the geometry and structure of the protein residue it represents is variable. For instance, backbone-only featurizations simply ignore the orientations of the side chain atoms in the residue, so the geometric GNN must account for this information implicitly.
> - However, the extent to which providing complete information about all atoms and side chain orientations is debatable, as the exact coordinates from PDB files are known to contain artifacts from structure determination via crystallography (see [ProteinMPNN](https://www.science.org/doi/10.1126/science.add2187)). This is also one of the interesting outcomes of our benchmark – letting the model implicitly learn about side chain orientation performs better or equally well as explicitly providing complete side chain atomic information.
>
> **Limitations of pre-training tasks:**
> - We only focussed on pre-training tasks that roughly fall under the category of corrupting information in the input (eg. sequence, coordinates) and tasking the model with producing the uncorrupted input. We were particularly interested in self-supervised objectives that were (1) extremely scalable, so as to pre-train on the large scale AlphaFold Protein Structure Database (AFDB) of 2.4M structures; and (2) train protein representations at the fine-grained node level, so as to be general-purpose across the downstream tasks considered.
> - We did not consider other possible tasks that have also been used in the literature, such as contrastive learning and generative modelling-inspired objectives. Most of these objectives are (1) computationally heavier and more cumbersome to set up than corruption-type objectives, making them harder to scale up, and (2) only train protein representations for the global/graph level and do not operate at the node level.
>
> ---
>
> > Would be beneficial to include a discussion about the generalizability of the benchmark results to the overall protein structure space, and how this translates to proteins not included in the current dataset.
>
> Indeed, we will elaborate on this further. By using the AFDB as a pre-training corpus, models trained using ProteinWorkshop should, in principle, exhibit strong generalisation to the currently known (and predicted) natural protein structure universe. The reason for this is that afdb_rep_v4, the FoldComp-curated clustering of the AFDB we used for pre-training, contains a non-redundant set of 2.4M structures, ie. each of the structures in the dataset has below 50% TM score with one another, indicating that these are unique folds. Any model trained on afdb_rep_v4 would have ‘seen’ any new natural protein fold in its pre-training corpus.
>
> ---
>
> > Missing a discussion about how geometric models may be improved to surpass sequence-based models.
>
> We ultimately believe that future protein representation learning models may blur the boundaries of sequence-based vs geometric/structure-based modelling. Concurrent works submitted to ICLR have already been to explore combining geometric graph neural networks with pre-trained protein language models for improved performance on specific downstream tasks, and we believe this trend will continue for the foreseeable future. Moreover, another avenue worth considering for future research on geometric models, one particularly suited to such models, is generative modeling-based pre-training objectives. Some early works have begun to demonstrate the usefulness for downstream tasks of incorporating diffusion-based objectives as a pre-training regime for geometric graph neural networks. As more flexible generative modeling frameworks emerge, we anticipate this trend will also continue, as we will likely see methods such as geometric flow matching begin to supersede diffusion pre-training by instead pre-training models to learnably interpolate between one relevant biomolecular data domain to another. Since our benchmark is built with modularity in mind, our work is poised to quickly take advantage of new generative modeling advances for enhanced geometric pre-training.

---

> ### Author Response · Authors · 2023-11-18
> **Response to Reviewer Reviewer W2RU (Part 2/n=2)**
>
> > Missing information about the ease of use of the tools, and details about the computational resources required for using the benchmark.
>
> We thank the reviewer for raising this point. We have added a discussion on usability and computational resources to the appendix (pending upload of revised paper). In short, for benchmarking, all models are trained on 80Gb NVIDIA A100 GPUs. Moreover, all baseline and finetuning results are performed using a single GPU while pre-training is performed using four GPUs.
>
> The modular design of our benchmark means it can be readily adapted into different workflows easily. Firstly, the benchmark is pip-installable from PyPI and contains several importable modules, such as dataloaders, featurisers and models, that can be imported into new projects. This will aid in standardising the datasets and workflows used in protein representation learning. Secondly, the benchmark serves as an easily extendable template, which users can fork and work directly in, thereby reducing implementation burden. Lastly, we provide a CLI that can be used to quickly run single experiments and hyperparameter sweeps with minimal development time overhead.
>
> ---
> > Missing (i) aggregation of methods for improving model performance; and (ii) computation of uncertainties in evaluations.
>
> (i) In order to extract the best performance out of models, it is natural that end-users of ProteinWorkshop may want to perform ensembling of the predictions of various methods. However, our goal with this paper was to rigorously compare architectural choices around featurization, GNN layers, and pre-training/auxiliary tasks in a fair manner, as a way of demonstrating the utility of ProteinWorkshop. We did not set out with the goal of obtaining SOTA performance on all datasets.
>
> (ii) Could the reviewer please clarify what they meant by uncertainties in evaluation? We did tune the model hyperparameters fairly across all models, and performed each experiment with multiple random seeds in order to obtain standard deviations across our results. This allows us to make statistically significant takeaways from our benchmarking experiments.
>
> ---
> > Can the work be adapted for other biological macromolecules beyond protein structures? ...
>
> - Yes, advances in geometric GNN modelling and methodology should, in principle, be adaptable and translate well to modelling biomolecular complexes among combinations of proteins, small molecules, nucleic acids, and other elements.
> - While the architectural details of the latest version of AlphaFold are not known to the public, all previous architectures which successfully model biomolecular complexes also represent these systems as geometric graphs with atoms/residues embedded as nodes in 3D Euclidean space. Hence, geometric GNNs are the natural architecture for representation learning across biomolecular systems.
> - We currently focus on protein representation learning because: (1) large scale datasets for self-supervised learning, as well as well-defined downstream tasks, are readily available and accepted by the community; and (2) we see protein representation learning as a fundamental or **foundational task**, improving upon which should also advance applications in protein + X complexes. For instance, pre-trained node embeddings from ProteinWorkshop models can readily be ported as initial features for protein + X models.
> - Comparatively, the scale of data for biomolecular complexes is significantly lesser, and there is less consensus among the community on how to perform evaluation; see [PoseCheck](https://arxiv.org/abs/2308.07413) and [PoseBusters](https://arxiv.org/abs/2308.05777) which was also referenced in the new AlphaFold version.
>
> We will include a discussion on biomolecular complexes in the revised version, too. Indeed, we envision the future of protein representation learning to consider both independent proteins as well as their interactions.

---

### Author Response · Authors · 2023-11-23
**Revised manuscript available**

We'd like to extend our thanks to the reviewers once again for all their valuable feedback and for their engagement with our responses. We have now uploaded our revised manuscript which contains additional results for several tasks (GO, PPI site prediction, Antibody Developability, Reaction Class prediction). We hope these will be of interest to the reviewers and we would ask the reviewers to kindly note the significant resources deployed this week to perform these baselines (over 125 days of NVIDIA A100 GPU time).

While our evaluations are mostly complete, we commit to obtaining a full complement of baselines by the camera-ready deadline if accepted. Due to the number of configurations tested, it was not possible for us to perform triplicates in the rebuttal period, but we are committed to doing so.

Furthermore, we have highlighted changes to the main text and supplementary material, labelling portions that address specific concerns and points raised by reviewers in the margins. We have also added additional discussions and restructuring to the appendix based on our discussions with each reviewer (Appendix B).

While there is not much time remaining, we would be very glad to respond to any last minute concerns or discussions.

Ultimately, we hope that the engineering, resources, and compute hours dedicated to developing ProteinWorkshop will prove useful to the protein representation learning community.

Warmly,

Authors of Submission #6130

---

### Meta-Review · Area_Chair_K82L · 2023-12-06

**Metareview:**

The paper presents a benchmark suite for appraising protein structure representation learning methods, incorporating pre-training methods, downstream tasks, and pre-training corpora for a systematic evaluation of the quality of learned embeddings and their application in downstream tasks.

This benchmark suite stands to be of considerable benefit to those engaged in protein presentation learning. While two reviewers have responded positively, one has given a slightly negative review. There are, however, some concerns that remain:
- The performance figures of the baselines reported in this paper don't quite match up to some of the existing methods, which could potentially limit the usage of this benchmark.
- Further time and results are necessary, particularly in the context of removing potential leakage and introducing new methods, such as non-structure-based models.

**Justification For Why Not Higher Score:**

As mentioned above, there are still some outstanding concerns that need to be addressed.

**Justification For Why Not Lower Score:**

This benchmark suite stands to be of considerable benefit to those engaged in protein presentation learning, even in its current state.

---

### Decision · Program_Chairs · 2024-01-16

Accept (poster)